# Efficient Hyper-parameter Optimization with Cubic Regularization

**Zhenqian Shen**[1*]    **Hansi Yang**[2*]    **Yong Li**[1]    **James Kwok**[2]    **Quanming Yao**[1†]

[1]Department of Electronic Engineering, Tsinghua University, Beijing, China
[2]Department of Computer Science and Engineering,
Hong Kong University of Science and Technology, Hong Kong SAR, China
{szg22@mails, liyong07@mail, qyaoaa@mail}.tsinghua.edu.cn
{hyangbw, jamesk}@cse.ust.hk

## Abstract

As hyper-parameters are ubiquitous and can significantly affect the model performance, hyper-parameter optimization is extremely important in machine learning. In this paper, we consider a sub-class of hyper-parameter optimization problems, where the hyper-gradients are not available. Such problems frequently appear when the performance metric is non-differentiable or the hyper-parameter is not continuous. However, existing algorithms, like Bayesian optimization and reinforcement learning, often get trapped in local optimals with poor performance. To address the above limitations, we propose to use cubic regularization to accelerate convergence and avoid saddle points. First, we adopt stochastic relaxation, which allows obtaining gradient and Hessian information without hyper-gradients. Then, we exploit the rich curvature information by cubic regularization. Theoretically, we prove that the proposed method can converge to approximate second-order stationary points, and the convergence is also guaranteed when the lower-level problem is inexactly solved. Experiments on synthetic and real-world data demonstrate the effectiveness of our proposed method.

## 1   Introduction

In a machine learning system, hyper-parameters are parameters that control the learning process. Well-known examples include learning rate, and choice of loss functions or regularizers. Hyper-parameters play a crucial role in determining the ultimate performance of machine learning models [1; 2; 3], affecting both the convergence of training algorithms and the generalization capabilities of the final models. Therefore, hyper-parameter optimization, as a problem of finding the optimal values for the hyper-parameters in a machine learning system, is fundamental and important.

Since manual hyper-parameter tuning can be time-consuming and often inefficient to find better hyper-parameters, there have been many works on hyper-parameter optimization to automatically search for optimal hyper-parameters. Numerous existing hyper-parameter optimization algorithms compute the hyper-gradient with respect to hyper-parameter and utilize it to update current hyper-parameter [4; 5; 6; 7]. However, computing the hyper-gradient requires the objectives to be differentiable, and the hyper-parameters to be optimized must be continuous. As such, hyper-gradients cannot be obtained in many important hyper-parameter optimization problems. For instance, in recommendation system and knowledge graph completion, non-differentiable evaluation metrics such as MRR and NDCG are commonly used [8; 9; 10], which makes it impossible to compute hyper-gradients. Moreover, many

---

[*]Equal contribution.
[†]Corresponding author.

37th Conference on Neural Information Processing Systems (NeurIPS 2023).

hyper-parameters in machine learning procedures are discrete (e.g. number of layers and hidden dimension size), where computing their hyper-gradient is impossible either.

To solve these problems without the use of hyper-gradients, existing methods often use some derivative-free optimization algorithms, e.g., random search [3], Bayesian optimization [11] or genetic programming [12]. However, these methods are often very time-consuming, and cannot find optimal hyper-parameters. Another type of method is based on stochastic relaxation [13; 14; 15], which constructs another differentiable objective from the original upper-level (i.e. the objective to find the hyper-parameters that have the best performance) objective, and use gradient-based optimization algorithms to optimize this new objective. This framework also covers methods based on reinforcement learning [16]. However, existing works under this framework only considers first-order information (i.e., gradient) of the new objective, which is still not sufficient enough as the upper-level objective can be very complex.

To address the limitations of existing works, in this paper, we propose a novel algorithm for hyper-parameter optimization that does not depend on hyper-gradients. Based on stochastic relaxation, we use cubic regularization [17] to optimize the relaxed objective and consider inexact solutions for the lower-level problems. Specially, our contributions can be summarized as follows:

- We propose to utilize cubic regularization based on stochastic relaxation to accelerate convergence and avoid saddle points in hyper-parameter optimization problems.

- We provide theoretical analysis of the proposed method, showing that the proposed method converge to approximate second-order stationary points with only inexactly solving lower level objectives.

- Experiments on synthetic data and diverse applications demonstrate that the proposed method can help find hyper-parameters that lead to models with good performances.

## 2 Related Works

### 2.1 Hyper-parameter Optimization

Hyper-parameter optimization [18; 19], as a subfield of automated machine learning (AutoML) [20], aims to automate the design and tuning of machine learning models. Most existing works on hyper-parameter optimization adopt the bi-level formulation [21; 4], where hyper-parameters are optimized in the upper-level objective, and model parameters are trained in the lower-level objective.

Generally, hyper-parameter optimization algorithms can be divided into two types. The first is derivative-free methods, which only depends on the objective value. For example, grid search [22] and random search [3] are two simple search methods to try different hyper-parameter values and choose the best among them based on the upper-level objective. Another example is model-based Bayesian optimization [11; 2], which builds a probabilistic model to map different hyper-parameters to their evaluation performance and updates this model along with the optimization process. Genetic programming [12] starts with a population of randomly sampled hyper-parameters, and evolves new generations of hyper-parameters by selecting top-performing ones and recombining them. Hyperband [23] improves upon random search by stopping training models with bad performances early before convergence. These methods may suffer from huge computation cost when the number of hyper-parameters is more than a few, and usually do not have a theoretical guarantee for convergence.

Another approach uses hyper-gradient to update hyper-parameters by gradient descent [4; 5; 6; 7]. Despite their solid theoretical analysis and good empirical performances, these methods require both continuous hyper-parameter and differentiable upper-level objective to compute the hyper-gradient. In contrast, our method handles the cases where hyper-gradient is not available, such as when the hyper-parameters are discrete or the objectives are non-differentiable. Moreover, exact computation of the hyper-gradient can be time-consuming, and is only feasible for small models.

### 2.2 Cubic Regularization

Cubic regularization [17] is a novel technique for general unconstrained and smooth optimization problems. The main idea is to construct a cubic polynomial from the original problem, and update the parameters by finding the minimizer of this polynomial in each iteration. Since the pioneering work [17] in this direction, many variants have been proposed to improve its performance. For

example, ARC [24] proposes to use an approximate global minimizer to reduce the computation cost of cubic regularization method. More recently, cubic regularization is utilized to accelerate convergence in stochastic non-convex optimization problems [25; 26; 27].

Compared to first-order optimization methods, cubic regularization can converge faster [17; 25] and escape saddle points [26]. Despite its solid theoretical foundations and good empirical performances, cubic regularization involves computing the full Hessian matrix, which prevents its application when the number of parameters is large (e.g., training a neural network). Nevertheless, since hyper-parameters are often low-dimensional, computing the Hessian matrix will not be a bottleneck. Our proposed method is also the first to use cubic regularization in a bi-level optimization problem.

## 3 Methodology

### 3.1 Problem Formulation

Mathematically, hyper-parameter optimization with stochastic relaxation [13] can be formulated as the following bi-level optimization problem:

$$\boldsymbol{\theta}^* = \arg\min_{\boldsymbol{\theta}} \left\{ \mathcal{J}(\boldsymbol{\theta}) = \mathbb{E}_{z \sim p_{\boldsymbol{\theta}}(z)}[H(w^*(z), z; D_{\text{val}})] \right\} \text{ s.t. } w^*(z) = \arg\min_{w} F(w, z; D_{\text{tra}}), \quad (1)$$

where $w$ denotes the model parameter and $z$ denotes the hyper-parameter. Through stochastic relaxation, the hyper-parameter $z$ is sampled from the probability distribution $p_{\boldsymbol{\theta}}(z)$ parameterized by $\boldsymbol{\theta}$. $H$ and $F$ are the performance evaluation metrics for the training dataset $D_{\text{tra}}$ and the validation dataset $D_{\text{val}}$, respectively.

Different from directly searching for good hyper-parameters, (1) introduces a probability distribution $p_{\boldsymbol{\theta}}(z)$ on hyper-parameter $z$, so that we can optimize $\boldsymbol{\theta}$ instead of $z$. The following Lemma shows that optimizing $\boldsymbol{\theta}$ is equivalent to finding the best hyper-parameter $z$ on the validation set:

**Lemma 3.1.** *Assume that for any $z^*$, there exists a $\boldsymbol{\theta}$ (could be infinite) such that $p_{\boldsymbol{\theta}}(z) = \delta(z - z^*)$, where $\delta(\cdot)$ is the Dirac's delta function. Then, (i) $\inf_{\boldsymbol{\theta}} \mathcal{J}(\boldsymbol{\theta}) = \min_z H(w^*(z), z; D_{val})$; (ii) The optimal $\bar{\boldsymbol{\theta}}$ in (1) satisfies $p_{\bar{\boldsymbol{\theta}}}(z) = \delta(z - \arg\min_z H(w^*(z), z; D_{val}))$.*

The proof is in Appendix A. The solution of (1) can be easier as $\mathcal{J}(\boldsymbol{\theta})$ in (1) is always differentiable, even when we cannot compute the hyper-gradient with respect to $z$. As is introduced in Section 1, hyper-parameter optimization problems where hyper-gradient cannot be computed are really common. Here are two examples:

*Example 1: Hyper-parameter optimization for knowledge graph completion [9; 28].* A knowledge graph $G$ is a set of triplets $\{(h, r, t)\}$ that indicates entity $h$ has relation $r$ to another entity $t$. Knowledge graph completion is to learn a model to predict unknown facts (i.e., triplets with missing components). Here, we mainly focus on embedding-based models, and we consider the task of tuning hyper-parameters for this type of model. Under the formulation of (1), $w$ refers to the embedding model parameters, $z$ refers to the hyper-parameters (e.g. learning rate, regularizer, scoring function). $H$ is a discrete evaluation metric (e.g. MRR, Hit@10). $F$ corresponds to the loss function for training. $D_{\text{tra}}$ and $D_{\text{val}}$ are formed by training and validation triplets in the knowledge graph.

*Example 2: Schedule search for learning with noisy labels [29; 30].* In this application, we have a noisy training dataset (i.e., labels may be incorrect), and we use a proportion schedule to control the proportion of small-loss samples for model training in each epoch, which can avoid over-fitting on incorrect labels. Existing works [29; 30] demonstrate that different schedules have significant impact on the final performance. Therefore, we consider using hyper-parameter $z$ to express this schedule, and we can optimize $z$ to help find good schedules that lead to good models. Under the formulation of (1), we let $w$ be the model parameter, $w^*(z)$ be the final model parameters trained with schedule parameterized by $z$. $F$ denotes the training loss and $H$ denotes the negative validation set accuracy. $D_{\text{tra}}$ is the training dataset with noisy label, and $D_{\text{val}}$ is the clean validation dataset.

### 3.2 Proposed Method

Since the upper-level objective in (1) can be highly non-convex, common optimization methods such as gradient descent and Newton method may get stuck in saddle points, which leads to poor

performance. As such, we propose to use cubic regularization, which can avoid convergence to saddle points, to optimize the relaxed objective $\mathcal{J}(\boldsymbol{\theta})$ in (1). We first briefly introduce the main procedure of using cubic regularization to minimize a given objective $\mathcal{J}(\boldsymbol{\theta})$. Denote $\boldsymbol{\theta}^m$ as the value of $\boldsymbol{\theta}$ in the $m$-th iteration, cubic regularization will first construct the following surrogate:

$$c_m(\boldsymbol{\Delta}) = \mathcal{J}(\boldsymbol{\theta}^m) + (\nabla \mathcal{J}(\boldsymbol{\theta}^m))^\top \boldsymbol{\Delta} + \frac{1}{2}\boldsymbol{\Delta}^\top (\nabla^2 \mathcal{J}(\boldsymbol{\theta}^m))\boldsymbol{\Delta} + \frac{\rho}{6}\|\boldsymbol{\Delta}\|_2^3, \tag{2}$$

where $\rho$ is a parameter set in advance, We will then update $\boldsymbol{\theta}^m$ by $\boldsymbol{\theta}^{m+1} = \boldsymbol{\theta}^m + \arg\min_{\boldsymbol{\Delta}} c_m(\boldsymbol{\Delta})$, which involves minimizing the above surrogate as a sub-procedure.

To compute gradient $\nabla \mathcal{J}(\boldsymbol{\theta}^m)$ and Hessian $\nabla^2 \mathcal{J}(\boldsymbol{\theta}^m)$, we have the following propositions:

**Proposition 3.2.** $\nabla \mathcal{J}(\boldsymbol{\theta}) = \mathbb{E}_{p_{\boldsymbol{\theta}}(z)}[s^*(\boldsymbol{\theta}; z)]$, where $s^*(\boldsymbol{\theta}; z) = H(w^*(z), z; D_{val}) \cdot \nabla \log p_{\boldsymbol{\theta}}(z)$.

**Proposition 3.3.** $\nabla^2 \mathcal{J}(\boldsymbol{\theta}) = \mathbb{E}_{p_{\boldsymbol{\theta}}(z)}[\boldsymbol{H}^*(\boldsymbol{\theta}; z)]$, where $\boldsymbol{H}^*(\boldsymbol{\theta}; z) = H(w^*(z), z; D_{val}) \cdot (\nabla \log p_{\boldsymbol{\theta}}(z) \nabla \log p_{\boldsymbol{\theta}}(z)^\top + \nabla^2 \log p_{\boldsymbol{\theta}}(z))$.

Proofs of these two propositions are in Appendix A. Note that we do not require the original upper-level objective $H$ to be differentiable, as we only need to use the value of upper-level objective $H(w^*(z), z; D_{\text{val}})$ to compute the gradient and Hessian of $\mathcal{J}(\boldsymbol{\theta})$. Nevertheless, these two propositions still cannot be directly used, as they require infinite number of samples $z$ as well as the corresponding exact lower-level solutions $w^*(z)$, and both of them are impossible to obtain in practice. As such, we first define $w'(z)$ to be an approximate solution of the lower-level objective with hyper-parameter $z$. Then we define $s(\boldsymbol{\theta}; z) = H(w'(z), z; D_{\text{val}}) \cdot \nabla \log p_{\boldsymbol{\theta}}(z)$ and $\boldsymbol{H}(\boldsymbol{\theta}; z) = H(w'(z), z; D_{\text{val}}) \cdot (\nabla \log p_{\boldsymbol{\theta}}(z) \nabla \log p_{\boldsymbol{\theta}}(z)^\top + \nabla^2 \log p_{\boldsymbol{\theta}}(z))$ as approximations of $s^*(\boldsymbol{\theta}; z)$ and $\boldsymbol{H}^*(\boldsymbol{\theta}; z)$. With these notations, the approximated gradient (denoted as $\boldsymbol{g}^m$) and Hessian (denoted as $\boldsymbol{B}^m$) in the $m$-th iteration are computed as:

$$\boldsymbol{g}^m = \frac{1}{K}\sum_{k=1}^K s(\boldsymbol{\theta}^m; z^k), \qquad (3) \qquad\qquad \boldsymbol{B}^m = \frac{1}{K}\sum_{k=1}^K \boldsymbol{H}(\boldsymbol{\theta}^m; z^k). \qquad (4)$$

In other words, we will draw $K$ different hyper-parameters $z^k$ from the distribution $p_{\boldsymbol{\theta}^m}(z)$ in $m$-th iteration, and use them to obtain the corresponding approximate solutions $w'(z^k)$. Then we can compute the approximate gradient $\boldsymbol{g}^m$ and Hessian $\boldsymbol{B}^m$, and an approximate surrogate $\tilde{c}_m(\boldsymbol{\theta})$ is then given by:

$$\tilde{c}_m(\boldsymbol{\Delta}) = (\boldsymbol{g}^m)^\top \boldsymbol{\Delta} + \frac{1}{2}\boldsymbol{\Delta}^\top \boldsymbol{B}^m \boldsymbol{\Delta} + \frac{\rho}{6}\|\boldsymbol{\Delta}\|_2^3. \tag{5}$$

Compared to $c_m(\boldsymbol{\theta})$ in (2), we remove the constant term $\mathcal{J}(\boldsymbol{\theta}^m)$, which will not affect the minimizer. $\boldsymbol{\theta}$ is then updated as $\boldsymbol{\theta}^{m+1} = \boldsymbol{\theta}^m + \boldsymbol{\Delta}^m$, where $\boldsymbol{\Delta}^m = \arg\min_{\boldsymbol{\Delta}} \tilde{c}_m(\boldsymbol{\Delta})$ is the minimizer of this new surrogate and can be obtained by existing solvers on cubic regularization (e.g., [24]). The complete algorithm is shown in Algorithm 1.

---

**Algorithm 1** Hyper-parameter optimization with cubic regularization.

---

1: Initialize $\boldsymbol{\theta}^0 = \mathbf{1}$ and $z^*$ randomly sampled from $p_{\boldsymbol{\theta}^0}(z)$.
2: **for** $m = 0, \ldots, M-1$ **do**
3:     **for** $k = 0, \ldots, K-1$ **do**
4:         Draw hyper-parameter $z^k$ from $p_{\boldsymbol{\theta}^m}(z)$;
5:         Optimize lower-level objective in (1) to obtain $w'(z^k)$; // *most expensive step*
6:         Update $z^* = z^k$ if we have a better validation performance with hyper-parameter $z^k$
7:     **end for**
8:     Compute $\boldsymbol{g}^m$ using (3) and $\boldsymbol{B}^m$ using (4);
9:     Compute $\boldsymbol{\Delta}^m$ by $\boldsymbol{\Delta}^m = \arg\min_{\boldsymbol{\Delta}} \tilde{c}_m(\boldsymbol{\Delta})$;
10:    Update $\boldsymbol{\theta}^{m+1} = \boldsymbol{\theta}^m + \boldsymbol{\Delta}^m$;
11: **end for**
12: Perform the training step with $z^*$ and obtain the final model parameter $w^*$;
13: **return** Final model parameter $w^*$

---

Since the number of hyper-parameters is often not too large, $\boldsymbol{\theta}$ is also a low-dimensional variable in most cases. For example, for the knowledge graph application in Section 4.2.1, the dimension of

$\boldsymbol{\theta}$ is only 25, as we sum up the number of possible values for each hyper-parameter. This is almost negligible compared with the embedding model, which often has millions of parameters. As such, optimizing (5) takes little time compared with solving the lower-level problem, which will also be empirically verified in Section 4.4.

### 3.3 Theoretical Analysis

Previous works that use cubic regularization for stochastic non-convex optimization [25; 26] do not consider bi-level optimization problems, and assume unbiased estimations of gradient and Hessian. As such, their analysis cannot be directly generalized here as our approximated gradient and Hessian are not unbiased due to inexact lower-level solutions. To analyze the convergence of Algorithm 1, we first introduce the definition for saddle point and approximate second-order stationary point, which is also used in [26] for convergence analysis:

**Definition 3.4** ([26]). For an objective $\mathcal{J}$ with $\rho$-Lipschitz Hessian, [3] $\boldsymbol{\theta}$ is an $\epsilon$-*second-order stationary point* of $\mathcal{J}$ if $\|\nabla \mathcal{J}(\boldsymbol{\theta})\|_2 \leq \epsilon$ and $\lambda_{\min}(\nabla^2 \mathcal{J}(\boldsymbol{\theta})) \geq -\sqrt{\rho\epsilon}$.

The definition of $\epsilon$-second-order stationary point can be regarded as a extension of $\epsilon$-first-order stationary point. For $\epsilon$-first-order stationary points, we only require the gradient $\nabla \mathcal{J}(\boldsymbol{\theta})$ to satisfy $\|\nabla \mathcal{J}(\boldsymbol{\theta})\|_2 \leq \epsilon$. And for $\epsilon$-second-order stationary points, we also require the eigenvalues of its Hessian to be large enough, so as to avoid getting stuck in saddle points as in Definition 3.5.

**Definition 3.5.** For an objective $\mathcal{J}$ with $\rho$-Lipschitz Hessian, $\boldsymbol{\theta}$ is an $\epsilon$-*saddle point* of $\mathcal{J}$ if $\|\nabla \mathcal{J}(\boldsymbol{\theta})\|_2 \leq \epsilon$ and $\lambda_{\min}(\nabla^2 \mathcal{J}(\boldsymbol{\theta})) < -\sqrt{\rho\epsilon}$.

Now we make the following assumptions. Assumption 3.6 (i) can be easily satisfied as the upper-level objective $H$ is naturally lower-bounded in real-world applications, and the parameter $\rho$ in (2) should match (at least not smaller than) the upper bound $\rho$ in Assumption 3.6 (i). This can be done by gradually increasing $\rho$ in (2) to ensure convergence. Assumption 3.6 (ii) is also satisfied by common probability distributions used in our experiments (e.g. sigmoid function for experiments on synthetic data and softmax liked distribution for hyper-parameter optimization for knowledge graph completion). Assumption 3.6 (i), (iii) and (iv) are popularly used in existing works on cubic regularization [25; 26].

**Assumption 3.6.** Following assumptions on $\mathcal{J}$ are made:

(i) Objective: $\inf_{\boldsymbol{\theta}} \mathcal{J}(\boldsymbol{\theta}) > -\infty$ and $\mathcal{J}$ is second-order differentiable with $\rho$-Lipschitz Hessian.

(ii) Bounded gradient and Hessian of probability distribution function $p_{\boldsymbol{\theta}}(z)$: $\|\nabla \log p_{\boldsymbol{\theta}}(z)\|_2 \leq Q_1$ and $\|\nabla^2 \log p_{\boldsymbol{\theta}}(z)\|_2 \leq Q_2$.

(iii) Bounded variance and bias in gradient estimation: $\mathbb{E}_{p_{\boldsymbol{\theta}}(z)}\big(\|\boldsymbol{s}(\boldsymbol{\theta}; z) - \mathbb{E}_{p_{\boldsymbol{\theta}}(z)}(\boldsymbol{s}(\boldsymbol{\theta}; z))\|_2^2\big) \leq \sigma_1^2$ and $\|\boldsymbol{s}(\boldsymbol{\theta}; z) - \mathbb{E}_{p_{\boldsymbol{\theta}}(z)}(\boldsymbol{s}(\boldsymbol{\theta}; z))\|_2 \leq M_1$ for all $\boldsymbol{\theta}$ and $z$.

(iv) Bounded error in Hessian estimation: denote the spectral norm of a matrix $\boldsymbol{A}$ (i.e., the maximum singular value of $\boldsymbol{A}$) as $\|\boldsymbol{A}\|_{\text{sp}}$, we have $\mathbb{E}_{p_{\boldsymbol{\theta}}(z)}\big(\|\boldsymbol{H}(\boldsymbol{\theta}; z) - \mathbb{E}_{p_{\boldsymbol{\theta}}(z)}(\boldsymbol{H}(\boldsymbol{\theta}; z))\|_{\text{sp}}\big) \leq \sigma_2$ and $\|\boldsymbol{H}(\boldsymbol{\theta}; z) - \mathbb{E}_{p_{\boldsymbol{\theta}}(z)}(\boldsymbol{H}(\boldsymbol{\theta}; z))\|_{\text{sp}} \leq M_2$ for all $\boldsymbol{\theta}$ and $z$.

The following theorem guarantees the convergence of Algorithm 1 (proof is in Appendix B).

**Theorem 3.7.** *For any $\mathcal{J}$ satisfying Assumption 3.6, and approximate lower-level solution $w'(z)$ with* $|H(w^*(z), z; D_{val}) - H(w'(z), z; D_{val})| \leq \min\big(\frac{\epsilon}{64Q_1}, \frac{\sqrt{\rho\epsilon}}{72(Q_1^2 + Q_2)}\big)$, *there exists $K = \bar{\mathcal{O}}\big(\frac{1}{\epsilon^2}\log(\frac{1}{\delta})\big)$, with probability $1 - \delta$ for any $\delta > 0$, such that Algorithm 1 produces an $\epsilon$-second-order stationary point $\boldsymbol{\theta}^M$ of $\mathcal{J}$, with $M = \mathcal{O}\big(\frac{\sqrt{\rho}(\mathcal{J}(\boldsymbol{\theta}^0) - \mathcal{J}(\boldsymbol{\theta}^*))}{\epsilon^{1.5}}\big)$.*

Theorem 3.7 shows that cubic regularization (Algorithm 1) takes only $M = \mathcal{O}(1/\epsilon^{1.5})$ iterations and $KM = \bar{\mathcal{O}}(1/\epsilon^{3.5})$ samples to obtain an $\epsilon$-second-order stationary point. This is strictly faster than gradient-based optimization methods [14; 15], which require $\bar{\mathcal{O}}(1/\epsilon^4)$ samples only to obtain an $\epsilon$-first-order stationary point. We also note that the convergence rate depends on two parts: inexactness

---

[3] In other words, for any $z, y$, we have $\|\nabla^2 \mathcal{J}(z) - \nabla^2 \mathcal{J}(y)\|_{\text{sp}} \leq \rho \|z - y\|_2$, where $\|\cdot\|_{\text{sp}}$ denotes the matrix spectral norm.

of lower-level solutions $|H(w^*(z), z; D_{\text{val}}) - H(w'(z), z; D_{\text{val}})|$, and the number of hyper-parameter samples $K$ to compute the approximate gradient $\boldsymbol{g}^m$ and Hessian $\boldsymbol{B}^m$. To better demonstrate their impacts, we consider the following two special cases:

- Set $K$ to $\infty$: in such case, the convergence is solely controlled by the inexactness of lower-level solutions. To obtain a good $\boldsymbol{\theta}$ (hence good hyper-parameter $z$), we need to obtain good lower-level solutions, which is intuitive as inaccurate lower-level solutions cannot help us identify good hyper-parameters.

- Using exact lower-level solutions: in such case, the convergence is solely controlled by the number of hyper-parameter samples $K$. To obtain a good $\boldsymbol{\theta}$ (hence good hyper-parameter $z$), we need $K$ to be sufficiently large to accurately compute the approximate gradient $\boldsymbol{g}^m$ and Hessian $\boldsymbol{B}^m$. This is also intuitive as inaccurate gradient or Hessian cannot lead to good $\boldsymbol{\theta}$, and matches previous analysis [25; 26] that focus on single-level optimization problems.

The above analysis will also be justified by empirical results in Section 4.3, where we consider the impact of different number of samples $K$ and different training epochs in lower-level, which causes the inexact solution of the lower-level objective.

**Comparison with existing works**   While there have been many works on hyper-parameter optimization [30; 14; 15] based on stochastic relaxation [13], our method has two significant differences. First, to the best of our knowledge, our method is the first method with convergence guarantee to approximate second-order stationary points (second-order convergence guarantees), while previous methods only have first-order convergence guarantees [30; 15]. As such, our method can avoid convergence to bad saddle points. While a recent work [31] also claims that they are able to escape saddle points in bi-level optimization, they require hyper-gradients to be available and more strict assumptions on objectives. Moreover, our method allows the use of inexact lower-level solutions, while existing methods all assume lower-level solutions are exact. Since lower-level problems are often complex non-convex problems, inexact lower-level solutions are more realistic and suitable for convergence analysis. Detailed comparison upon existing works can also be found in Appendix C.

## 4   Experiments

### 4.1   Experiment on Synthetic Data

First, we consider a problem of filtering useful features for a linear model [32; 33]. We construct a synthetic dataset on 5-way classification, where the inputs are 50-dimensional vectors. Of the 50 features, 25 have different distributions based on their ground-truth labels, generated from Gaussian distributions with different means for each class and the same variance. The remaining 25 features are filled with i.i.d. Gaussian white noise. The hyper-parameter $z \in \{0, 1\}^{50}$ is a 50-dimensional 0/1 vector to specify which dimensions are masked. We use cross entropy loss as the loss function in the lower-level objective and use AUC to measure the performance of the classifier in the upper-level objective. For the mask of $i$-th dimension $z_i$, we use sigmoid function to represent the probability to mask that dimension, i.e., $p_{\theta_i}(z_i = 1) = 1/(1+\exp(-\theta_i))$ and $p_{\theta_i}(z_i = 0) = 1 - p_{\theta_i}(z_i = 1)$. The complete distribution $p_{\boldsymbol{\theta}}(z)$ is represented by successive multiplications $p_{\boldsymbol{\theta}}(z) = \prod_{i=1}^{50} p_{\theta_i}(z_i)$. Experiments are conducted on a 24GB NVIDIA GeForce RTX 3090 GPU.

We compare the proposed method (*Cubic*) with methods that are also based on stochastic relaxation, including gradient descent (*GD*) [34], natural gradient (*NG*) [35], and Newton's methods (*Newton*) [36]. The validation AUC of different methods are shown in Figure 1(a). We can see that our proposed method clearly outperforms other methods based on stochastic relaxation.

Furthermore, to empirically verify that our proposed method does escape bad saddle points, we compare the eigenvalues of Hessian matrix $\nabla^2 \mathcal{J}(\boldsymbol{\theta})$ with $\boldsymbol{\theta}$ found by different methods. For easier comparison, we divide each method's eigenvalues by their largest, which makes different methods have the same largest eigenvalue as 1. As can be seen from Figure 1(b), only our proposed method obtains an optimized $\boldsymbol{\theta}$ that make the negative eigenvalues less significant compared to positive eigenvalues, which means only our proposed method can better escape the saddle points in $\mathcal{J}(\boldsymbol{\theta})$.

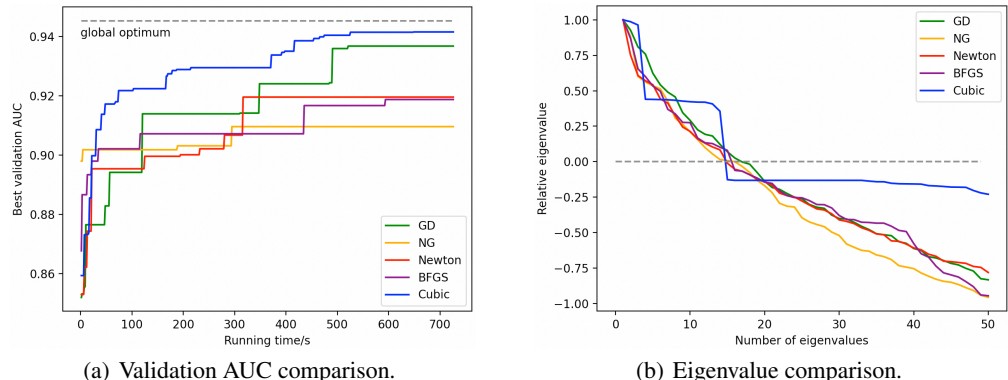

(a) Validation AUC comparison.          (b) Eigenvalue comparison.

Figure 1: Comparison among different gradient based methods for mask learning in synthetic data classification.

## 4.2 Experiments on Real-world Data

In this section, we mainly consider two applications, hyper-parameter optimization for knowledge graph completion and schedule search on learning with noisy training data, which are both introduced as examples in Section 3.1. For baselines, we choose hyper-parameter optimization methods that are commonly used in hyper-parameter optimization literature [19; 30; 28], including random search (*Random*) [3], Bayesian optimization (*BO*) [11], Hyperband [23] and reinforcement learning (*RL*) [37]. These baseline methods are applied to the original hyper-parameter search problem instead of the problem after stochastic relaxation.

### 4.2.1 Hyper-parameter Optimization for Knowledge Graph Completion

For this application, we need to tune several discrete hyper-parameters for a knowledge graph embedding model. The hyper-parameters to be tuned include negative sampling number, regularizer, loss function, gamma (for margin ranking loss), initializer, learning rate and score function, and more details can be found in Appendix E.2.

As all hyper-parameters considered in this application are discrete, we choose softmax-liked distributions to represent the probability to select a specific value for each hyper-parameter. For example, consider a hyper-parameter $z_i$ with possible values $\{z_{i1}, ..., z_{in}\}$, the probability of $z_i = z_{ij}$ is $p(z_i = z_{ij}) = \frac{\exp(\theta_{ij})}{\sum_{k=1}^{n} \exp(\theta_{ik})}$, where $\theta_{ij}$ denote the probability function parameter corresponding to hyper-parameter $z_{ij}$. The dimension of distribution parameter is 25. Two well-known knowledge graph datasets, FB15k237 [38] and WN18RR [39], are used in experiments and their statistics are in Appendix E.2.

The experimental results are presented in Figure 2, which shows the achieved maximum reciprocal rank (MRR) [4] with respect to the number of trained models for different search algorithms. Our proposed method exhibits rapid convergence, as Theorem 3.7 guarantees its fast convergence rate. Moreover, our proposed method outperforms other search algorithms and is more close to the global optimum in terms of MRR because cubic regularization allows our proposed method to escape from bad saddle points, while other methods may get trapped in such saddle points.

### 4.2.2 Schedule Search on Learning with Noisy Training Data

For this application, we consider the problem of searching for a schedule $R_z(t)$ parameterized by hyper-parameter $z$ [30]. The hyper-parameter $z$ is divided into two parts: $z \equiv (\boldsymbol{\alpha}, \{\boldsymbol{\beta}^i\})$, and $R_z(t)$ is parameterized as follows: $R_z(t) \equiv \sum_{i=1}^{I} \alpha_i r^i(t; \boldsymbol{\beta}^i)$, where $\boldsymbol{\alpha} = (\alpha_1, \dots, \alpha_I)$ can be seen as a set of weights with $\alpha_i \geq 0$ and $\sum_{i=1}^{I} \alpha_i = 1$. Settings of functions $r^i(t; \boldsymbol{\beta}^i)$ parameterized with $\boldsymbol{\beta}^i$ are shown in Table 1. Here the dimension of hyper-parameter is 36.

---

[4]Denote the number of test triplets as $n$, and the rank of $m$th triplet prediction as $\text{rank}_m$, MRR $= \frac{1}{n} \sum_{m=1}^{n} \frac{1}{\text{rank}_m}$.

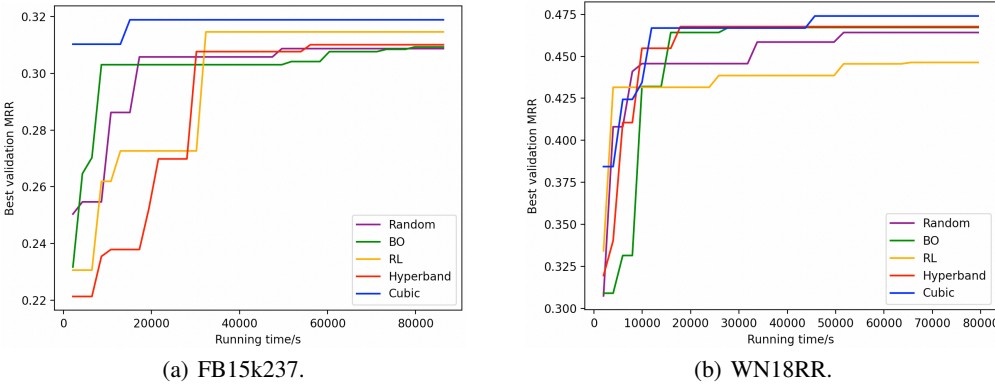

(a) FB15k237.  (b) WN18RR.

Figure 2: Validation MRR w.r.t. total number of trained models during search in hyper-parameter optimization for knowledge graph completion.

We use CIFAR-10 dataset for experiments with 50k, 5k, 5k image samples as training, validation, test data, respectively. We consider two different label noise settings: 50% symmetric noise and 45% pair flipping noise, and detailed generation process for the two settings is introduced in Appendix E.3.

Figure 3 shows the validation accuracy with respect to the total number of trained models in the search process. As can be seen, our method with cubic regularization is more efficient than the other algorithms compared. Moreover, our proposed method converges to a more accurate model because of the ability of cubic regularization to escape from bad saddle points.

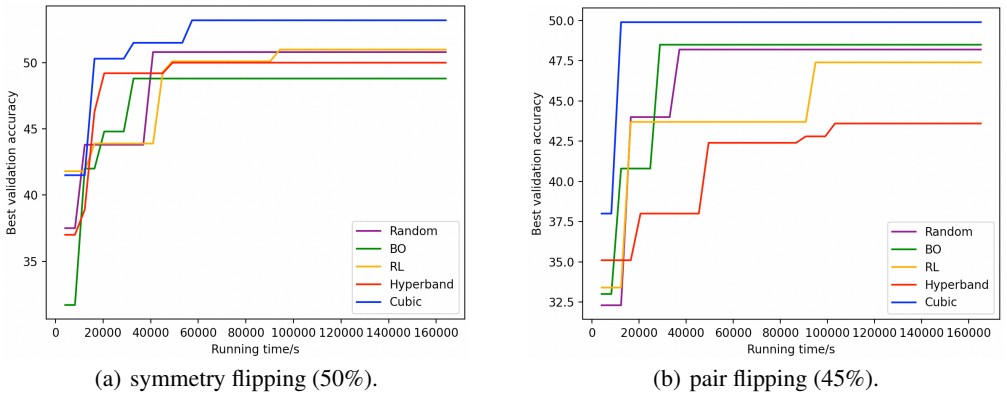

(a) symmetry flipping (50%).  (b) pair flipping (45%).

Figure 3: Validation accuracy w.r.t. total number of trained models during search for schedule search on learning with noisy training data.

### 4.3 Ablation Study

Here we conduct ablation study on two important factors of our proposed method: the number of samples $K$ (which affects the error in estimating the gradient and Hessian of $\mathcal{J}(\boldsymbol{\theta})$) and number of training epochs for each model (which causes the inexact solution of the lower-level objective). The experiments are done on the synthetic data experiment in section 4.1. Results are shown in Figure 4, where in the first row we fix the number of upper-level iterations among different parameter settings, while overall training time is controlled the same in the second row. We fix the epoch number as 40 for experiment on $K$ and set $K = 5$ for

Table 1: The basis functions used to define the search space in the experiments. Here, $T$ is the total number of training epochs.

| $r^1(t;\boldsymbol{\beta}^1)$ | $e^{-\beta_2^1 t^{\beta_1^1}} + \beta_3^1(\frac{t}{T})^{\beta_4^1}$ |
|---|---|
| $r^2(t;\boldsymbol{\beta}^2)$ | $e^{-\beta_2^2 t^{\beta_1^2}} + \beta_3^2 \frac{\log(1+t^{\beta_4^2})}{\log(1+T^{\beta_4^2})}$ |
| $r^3(t;\boldsymbol{\beta}^3)$ | $\frac{1}{(1+\beta_2^3 t)^{\beta_1^3}} + \beta_3^3(\frac{t}{T})^{\beta_4^3}$ |
| $r^4(t;\boldsymbol{\beta}^4)$ | $\frac{1}{(1+\beta_2^4 t)^{\beta_1^4}} + \beta_3^4 \frac{\log(1+t^{\beta_4^4})}{\log(1+T^{\beta_4^4})}$ |

experiment on epoch number. In the experiment on the number of samples $K$ (Figure 4(a)), we can see that our method performs better when $K$ is larger, which means that when the estimation for gradient and Hessian of $\mathcal{J}(\boldsymbol{\theta})$ is the more precise, the optimization of $\mathcal{J}(\boldsymbol{\theta})$ is more effective. However, when the overall training time is limited, excessively large $K$ (e.g. $K = 50$) causes decline

in performance because the number of parameter update for $\boldsymbol{\theta}$ is largely reduced. In the experiment for training epoch number (Figure 4(b)), we can see that the performance is better when the epoch number is larger (the solution of lower-level objective is more precise).

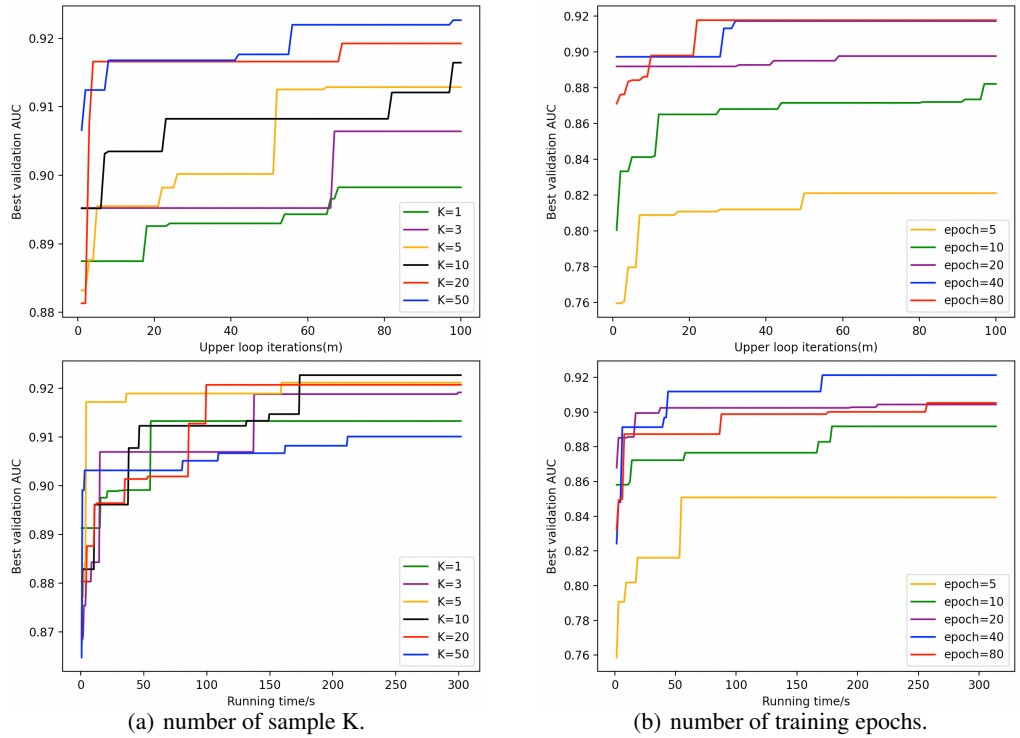

(a) number of sample K.  (b) number of training epochs.

Figure 4: Ablation study for our proposed method on synthetic data.

## 4.4 Time Cost in Experiments

In this section, we evaluate the time cost on different parts of our method in three experiments above. The total time cost of these hyper-parameter optimization problems can be mainly divided into two parts: (i) the model training with a specific hyper-parameter, and (ii) the update step of $\boldsymbol{\theta}$. Table 2 shows the the clock time in different datasets, where we conduct each experiment for 5 times. As we can see, time cost in model training is consistently far more than time cost in update step of $\boldsymbol{\theta}$, which shows that it is worthwhile to spend more time to exploit the curvature of upper-level optimization objective in (1).

Table 2: Clock time (in seconds) for model training and update step of $\boldsymbol{\theta}$.

|  | synthetic data (Section 4.1) | FB15k237 (Section 4.2.1) | WN18RR (Section 4.2.1) | CIFAR10 (Section 4.2.2) |
|---|---|---|---|---|
| model training | $710 \pm 37$ | $86298 \pm 760$ | $79471 \pm 936$ | $163992 \pm 801$ |
| update step of $\boldsymbol{\theta}$ | $16.1 \pm 4.2$ | $79.1 \pm 2.7$ | $72.4 \pm 2.3$ | $14.5 \pm 1.4$ |

## 5 Conclusion

In this paper, we address the challenge of hyper-parameter optimization in scenarios where hyper-gradients are unavailable. To tackle this problem, we introduce a novel algorithm that leverages cubic regularization based on stochastic relaxation. Our proposed method offers several advantages over existing approaches, including accelerated convergence and enhanced capability to escape from undesirable saddle points. Extensive experiments conducted on both synthetic and real-world datasets demonstrate the effectiveness of our proposed method in solving a wide range of hyper-parameter optimization problems.

## Acknowledgments

This work was supported in part by The National Key Research and Development Program of China under grant 2020AAA0106000, a grant from the Research Grants Council of the Hong Kong Special Administrative Region, China (Project No. HKU C7004-22G), and NSF of China (No. 92270106).

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
