# A Proof

## A.1 Lemma 3.1

*Proof.* Let $z^* = \arg\min_z H(w^*(z), z; D_{\text{val}})$. Obviously, we have

$$H(w^*(z), z; D_{\text{val}}) \geq H(w^*(z^*), z^*; D_{\text{val}}).$$

Recall that $\mathcal{J}(\boldsymbol{\theta}) = \int_{z \in \mathcal{S}} H(w^*(z), z; D_{\text{val}}) p_{\boldsymbol{\theta}}(z) \, dz$, then

$$\mathcal{J}(\boldsymbol{\theta}) \geq \int_{z \in \mathcal{S}} \left[ \min_z H(w^*(z), z; D_{\text{val}}) \right] p_{\boldsymbol{\theta}}(z) \, dz,$$

$$= \min_z H(w^*(z), z; D_{\text{val}}) \int_{z \in \mathcal{S}} p_{\boldsymbol{\theta}}(z) \, dz = \min_z H(w^*(z), z; D_{\text{val}}),$$

The equality in the first row holds only when $H(w^*(z), z; D_{\text{val}})$ is a constant as $H(w^*(z^*), z^*; D_{\text{val}})$ or $p_{\boldsymbol{\theta}}(z) = \delta(z - z^*)$. Therefore, $\inf_{\boldsymbol{\theta}} \mathcal{J}(\boldsymbol{\theta}) = \min_z H(w^*(z), z; D_{\text{val}})$. $\qquad\square$

## A.2 Proposition 3.2

*Proof.* By definition, we have:

$$\nabla \mathcal{J}(\boldsymbol{\theta}) = \int H(w^*(z), z; D_{\text{val}}) \nabla p_{\boldsymbol{\theta}}(z) dz$$

$$= \int H(w^*(z), z; D_{\text{val}}) \frac{\nabla p_{\boldsymbol{\theta}}(z)}{p_{\boldsymbol{\theta}}(z)} p_{\boldsymbol{\theta}}(z) dz$$

$$= \int H(w^*(z), z; D_{\text{val}}) \cdot \nabla \log(p_{\boldsymbol{\theta}}(z)) p_{\boldsymbol{\theta}}(z) dz$$

$$= \mathbb{E}_{p_{\boldsymbol{\theta}}} \left[ H(w^*(z), z; D_{\text{val}}) \cdot \nabla \log(p_{\boldsymbol{\theta}}(z)) \right]$$

which concludes our proof. $\qquad\square$

## A.3 Proposition 3.3

*Proof.* By definition, we have

$$\nabla^2 \mathcal{J}(\boldsymbol{\theta}) = \int H(w^*(z), z; D_{\text{val}}) \nabla^2 p_{\boldsymbol{\theta}}(z) dz = \mathbb{E}_{p_{\boldsymbol{\theta}}} \left[ H(w^*(z), z; D_{\text{val}}) \frac{\nabla^2 p_{\boldsymbol{\theta}}(z)}{p_{\boldsymbol{\theta}}(z)} \right]$$

Then we note that

$$\nabla^2 \log p_{\boldsymbol{\theta}}(z) = \nabla \left( \frac{\nabla p_{\boldsymbol{\theta}}(z)}{p_{\boldsymbol{\theta}}(z)} \right) = \frac{\nabla^2 p_{\boldsymbol{\theta}}(z)}{p_{\boldsymbol{\theta}}(z)} - \frac{\nabla p_{\boldsymbol{\theta}}(z) \nabla p_{\boldsymbol{\theta}}(z)^\top}{p_{\boldsymbol{\theta}}^2(z)}.$$

Thus we have

$$\frac{\nabla^2 p_{\boldsymbol{\theta}}(z)}{p_{\boldsymbol{\theta}}(z)} = \nabla^2 \log p_{\boldsymbol{\theta}}(z) + \frac{\nabla p_{\boldsymbol{\theta}}(z) \nabla p_{\boldsymbol{\theta}}(z)^\top}{p_{\boldsymbol{\theta}}^2(z)} = \nabla^2 \log p_{\boldsymbol{\theta}}(z) + \left( \frac{\nabla p_{\boldsymbol{\theta}}(z)}{p_{\boldsymbol{\theta}}(z)} \right) \left( \frac{\nabla p_{\boldsymbol{\theta}}(z)}{p_{\boldsymbol{\theta}}(z)} \right)^\top$$

$$= \nabla^2 \log p_{\boldsymbol{\theta}}(z) + (\nabla \log p_{\boldsymbol{\theta}}(z))(\nabla \log p_{\boldsymbol{\theta}}(z))^\top,$$

which concludes our proof. $\qquad\square$

# B Proofs for Cubic Algorithm

We first introduce some useful Lemmas in Appendix B.1, then prove Theorem 3.7 in Appendix B.2.

## B.1 Preliminary Lemmas

**Lemma B.1.** *[17] For a second-order differentiable function $f$ whose Hessian is $\rho$-Lipschitz continuous, i.e. $\left\|\nabla^2 f(z) - \nabla^2 f(y)\right\|_{sp} \le \rho \left\|z - y\right\|_2, \forall z, y$, then for all $z, y$ in the domain of $f$, we will have:*

$$\left\|\nabla f(y) - \nabla f(z) - \nabla^2 f(z)(y - z)\right\|_2 \le \frac{\rho}{2} \left\|y - z\right\|_2^2, \tag{6}$$

$$f(y) \le f(z) + (\nabla f(z))^\top (y - z) + \frac{1}{2}(y - z)^\top \nabla^2 f(z)(y - z) + \frac{\rho}{6} \left\|y - z\right\|_2^3. \tag{7}$$

**Lemma B.2.** *(Bernstein inequality [40]) For i.i.d. random vectors $\mathbf{v}_1, \ldots, \mathbf{v}_K \in \mathbb{R}^d$ with mean $\mathbb{E}[\mathbf{v}]$, if there exists a $M > 0$ such that $\|\mathbf{v}_i - \mathbb{E}[\mathbf{v}]\|_2 \le M$ holds for all $i = 1, \ldots, K$, then we will have*

$$P\left(\left\|{}^1\!/\!_K \sum\nolimits_{i=1}^{K} \mathbf{v}_i - \mathbb{E}[\mathbf{v}]\right\|_2 \ge t\right) \le 2d \cdot \exp\left(-\frac{Kt^2/2}{{}^1\!/\!_K \sum_{i=1}^{K} \mathrm{Var}(\mathbf{v}_i) + Mt/3}\right),$$

*for $t > 0$ where $\mathrm{Var}(\mathbf{v}_i) = \mathbb{E}(\|\mathbf{v}_i - \mathbb{E}[\mathbf{v}]\|_2^2)$, i.e., the variance of $\mathbf{v}_i$.*

**Lemma B.3.** *For any $c_1 > 0$ and $0 < \delta_1 < 1$, there exists a constant $K = \mathcal{O}\left({}^1\!/\!_{\epsilon^2} \log({}^1\!/\!_{\delta_1})\right)$, such that the sampled gradient $\mathbf{g}^m$ in (3) satisfies $\|\mathbf{g}^m - \nabla \mathcal{J}(\boldsymbol{\theta}^m)\|_2 \le c_1 \epsilon$, with probability at least $1 - \delta_1$.*

*Proof.* From Proposition 3.2, we have $\nabla \mathcal{J}(\boldsymbol{\theta}) = \mathbb{E}_{p_{\boldsymbol{\theta}}(z)}\left[\mathbf{s}^*(\boldsymbol{\theta}; z)\right]$, and $\mathbf{g}^m$ is computed by $\mathbf{g}^m = \frac{1}{K} \sum_{k=1}^{K} \mathbf{s}(\boldsymbol{\theta}^m; z^k)$, where $z^k$ is sampled from the probability distribution $p_{\boldsymbol{\theta}^m}(z)$. Then we have

$$
\begin{aligned}
\|\mathbb{E}_{p_{\boldsymbol{\theta}^m}(z)}(\mathbf{g}^m) - \nabla \mathcal{J}(\boldsymbol{\theta}^m)\|_2 &= \|\mathbb{E}_{p_{\boldsymbol{\theta}^m}(z)}((H(w'(z), z; D_{\mathrm{val}}) - H(w^*(z), z; D_{\mathrm{val}})) \cdot \nabla \log p_{\boldsymbol{\theta}^m}(z))\|_2 \\
&\le \min\left(\frac{\epsilon}{64Q_1}, \frac{\sqrt{\rho\epsilon}}{72(Q_1^2 + Q_2)}\right) \cdot \|\mathbb{E}_{p_{\boldsymbol{\theta}^m}(z)}(\nabla \log p_{\boldsymbol{\theta}^m}(z))\|_2 \\
&\le \frac{\epsilon}{64Q_1} \cdot \|\mathbb{E}_{p_{\boldsymbol{\theta}^m}(z)}(\nabla \log p_{\boldsymbol{\theta}^m}(z))\|_2 \le \frac{\epsilon}{64}
\end{aligned}
$$

where the last two steps use Assumption 3.6 (iv) and prerequisite of Theorem 3.7.

Since $\mathbb{E}_{p_{\boldsymbol{\theta}^m}(z)}\left(\|\mathbf{s}(\boldsymbol{\theta}^m; z) - \mathbb{E}_{p_{\boldsymbol{\theta}^m}(z)}(\mathbf{s}(\boldsymbol{\theta}^m; z))\|_2^2\right) \le \sigma_1^2$, we can directly obtain that:

$$\frac{1}{K} \sum\nolimits_{k=1}^{K} \mathrm{Var}(\mathbf{s}(\boldsymbol{\theta}^m, z^k)) \le \sigma_1^2$$

Then, for $\mathbf{g}^m$, by letting $t = b_1\epsilon$ in the Bernstein's inequality (Lemma B.2), we should have

$$P\left(\|\mathbf{g}^m - \mathbb{E}_{p_{\boldsymbol{\theta}}(z)}(\mathbf{g}^m)\|_2 \ge b_1\epsilon\right) \le 2d \exp\left(-\frac{Kb_1^2\epsilon^2/2}{{}^1\!/\!_K \sum_{k=1}^{K} \mathrm{Var}(\mathbf{s}(\boldsymbol{\theta}^m, z_k)) + b_1 M_1 \epsilon/3}\right).$$

Next, we have the following bound for R.H.S of the above inequality:

$$\frac{Kb_1^2\epsilon^2/2}{{}^1\!/\!_K \sum_{k=1}^{K} \mathrm{Var}(\mathbf{s}(\boldsymbol{\theta}^m, z^k)) + b_1 M_1 \epsilon/3} \ge \frac{Kb_1^2\epsilon^2/2}{\sigma_1^2 + b_1 M_1 \epsilon/3} = \frac{K}{4} \frac{2}{\sigma_1^2/b_1^2\epsilon^2 + M_1/3b_1\epsilon}.$$

Note that the last term has the harmonic mean of ${}^{b_1^2\epsilon^2}\!/\!_{\sigma_1^2}$ and ${}^{3b_1\epsilon}\!/\!_{M_1}$, then by the property of harmonic means, we should have:

$$\frac{K}{4} \frac{2}{\sigma_1^2/b_1^2\epsilon^2 + M_1/3b_1\epsilon} \ge \frac{K}{4} \min\left\{{}^{b_1^2\epsilon^2}\!/\!_{\sigma_1^2}, {}^{3b_1\epsilon}\!/\!_{M_1}\right\},$$

and this will give us

$$P\left(\left\|\mathbf{g}^m - \mathbb{E}_{p_{\boldsymbol{\theta}}(z)}(\mathbf{g}^m)\right\|_2 \ge b_1\epsilon\right) \le 2d \exp\left(-\frac{K}{4} \min\left\{{}^{b_1^2\epsilon^2}\!/\!_{\sigma_1^2}, {}^{3b_1\epsilon}\!/\!_{M_1}\right\}\right).$$

Then we let

$$\delta_1 = 2d \exp\left(-\frac{K}{4} \min\left\{b_1^2 \epsilon^2 / \sigma_1^2, 3b_1 \epsilon / M_1\right\}\right),$$

$$\text{i.e. } K = 4 \max\left\{M_1 / 3b_1 \epsilon, \sigma_1^2 / b_1^2 \epsilon^2\right\} \log(2d/\delta_1) = \mathcal{O}\left(1/\epsilon^2 \log(1/\delta_1)\right)$$

so that we can obtain $P(\|\boldsymbol{g}^m - \mathbb{E}_{p_{\boldsymbol{\theta}}(z)}(\boldsymbol{g}^m)\|_2 \geq b_1 \epsilon) \leq \delta_1$, which means $\|\boldsymbol{g}^m - \mathbb{E}_{p_{\boldsymbol{\theta}}(z)}(\boldsymbol{g}^m)\|_2 \leq b_1 \epsilon$ hold with probability at least $1 - \delta_1$.

Finally, letting $c_1 = \frac{1}{64} + b_1$, we can obtain that

$$\|\boldsymbol{g}^m - \nabla \mathcal{J}(\boldsymbol{\theta}^m)\|_2 \leq \|\mathbb{E}_{p_{\boldsymbol{\theta}}(z)}(\boldsymbol{g}^m) - \nabla \mathcal{J}(\boldsymbol{\theta}^m)\|_2 + \|\boldsymbol{g}^m - \mathbb{E}_{p_{\boldsymbol{\theta}}(z)}(\boldsymbol{g}^m)\|_2$$

$$\leq (\frac{1}{64} + b_1) \cdot \epsilon = c_1 \epsilon$$

holds with probability at least $1 - \delta_1$, which concludes our proof. $\qquad\square$

**Lemma B.4.** *(Bernstein inequality for matrix [40]) For i.i.d. random matrices $\boldsymbol{A}_1, \ldots, \boldsymbol{A}_K \in \mathbb{R}^{d \times d}$ with mean $\mathbb{E}[\boldsymbol{A}]$, if there exists a $M > 0$ such that $\|\boldsymbol{A}_i - \mathbb{E}[\boldsymbol{A}]\|_{sp} \leq M$ holds for all $i = 1, \ldots, K$, then we will have*

$$P\left(\left\|\frac{1}{K}\sum_{i=1}^{K} \boldsymbol{A}_i - \mathbb{E}[\boldsymbol{A}]\right\|_{sp} \geq t\right) \leq 2d^2 \cdot \exp\left(-\frac{Kt^2/2}{1/K \sum_{i=1}^{K} \mathrm{Var}(\boldsymbol{A}_i) + Mt/3}\right)$$

*for $t > 0$ where $\mathrm{Var}(\boldsymbol{A}_i) = \mathbb{E}(\|\boldsymbol{A}_i - \mathbb{E}[\boldsymbol{A}]\|_{sp}^2)$.*

**Lemma B.5.** *For any $c_2 \geq 0$ and $0 < \delta_2 < 1$, there exists a constant $K = \mathcal{O}\left(1/\epsilon^2 \log(1/\delta_1)\right)$, such that the sampled Hessian $\boldsymbol{B}^m$ satisfies:*

$$\forall \boldsymbol{v} \in \mathbb{R}^d, \quad \|(\boldsymbol{B}^m - \nabla^2 \mathcal{J}(\boldsymbol{\theta}^m))\boldsymbol{v}\|_2 \leq c_2 \sqrt{\rho \epsilon} \|\boldsymbol{v}\|_2,$$

*with probability at least $1 - \delta_2$.*

*Proof.* By the definition of spectral norm, we have

$$\forall \boldsymbol{v} \in \mathbb{R}^d, \|(\boldsymbol{B}^m - \nabla^2 \mathcal{J}(\boldsymbol{\theta}^m))\boldsymbol{v}\|_2 \leq c_2 \sqrt{\rho \epsilon} \|\boldsymbol{v}\|_2 \iff \|\boldsymbol{B}^m - \nabla^2 \mathcal{J}(\boldsymbol{\theta}^m)\|_{sp} \leq c_2 \sqrt{\rho \epsilon}$$

From Proposition 3.3, we have $\nabla^2 \mathcal{J}(\boldsymbol{\theta}) = \mathbb{E}_{p_{\boldsymbol{\theta}}(z)}\left[\boldsymbol{H}^*(\boldsymbol{\theta}; z)\right]$, and $\boldsymbol{B}^m$ is computed by $\boldsymbol{B}^m = \frac{1}{K}\sum_{k=1}^{K} \boldsymbol{H}(\boldsymbol{\theta}^m; z^k)$, where $z^k$ is sampled from the probability distribution $p_{\boldsymbol{\theta}^m}(z)$. Then we have

$$\|\mathbb{E}_{p_{\boldsymbol{\theta}}(z)}(\boldsymbol{B}^m) - \nabla^2 \mathcal{J}(\boldsymbol{\theta}^m)\|_{sp}$$

$$= \|\mathbb{E}_{p_{\boldsymbol{\theta}}(\boldsymbol{z})}((H(w^*(z), z; D_{\mathrm{val}}) - H(w'(z), z; D_{\mathrm{val}}))(\nabla \log p_{\boldsymbol{\theta}}(z) \nabla \log p_{\boldsymbol{\theta}}(z)^\top + \nabla^2 \log p_{\boldsymbol{\theta}}(z)))\|_{sp}$$

$$\leq \min\left(\frac{\epsilon}{64 Q_1}, \frac{\sqrt{\rho \epsilon}}{72(Q_1^2 + Q_2)}\right) \cdot \|\mathbb{E}_{p_{\boldsymbol{\theta}}(\boldsymbol{z})}(\nabla \log p_{\boldsymbol{\theta}}(z) \nabla \log p_{\boldsymbol{\theta}}(z)^\top + \nabla^2 \log p_{\boldsymbol{\theta}}(z))\|_{sp}$$

$$\leq \frac{\sqrt{\rho \epsilon}}{72(Q_1^2 + Q_2)} \cdot \|\mathbb{E}_{p_{\boldsymbol{\theta}}(\boldsymbol{z})}(\nabla \log p_{\boldsymbol{\theta}}(z) \nabla \log p_{\boldsymbol{\theta}}(z)^\top + \nabla^2 \log p_{\boldsymbol{\theta}}(z))\|_{sp}$$

$$\leq \frac{\sqrt{\rho \epsilon}}{72(Q_1^2 + Q_2)} \cdot \mathbb{E}_{p_{\boldsymbol{\theta}}(\boldsymbol{z})}(\|\nabla \log p_{\boldsymbol{\theta}}(z) \nabla \log p_{\boldsymbol{\theta}}(z)^\top\|_{sp} + \|\nabla^2 \log p_{\boldsymbol{\theta}}(z)\|_{sp})$$

$$\leq \frac{\sqrt{\rho \epsilon}}{72}$$

where in the first and the last inequation, we use Assumption 3.6 (iv) and prerequisite of Theorem 3.7, relatively.

With $\mathbb{E}_{p_{\boldsymbol{\theta}}(z)}\left(\|\boldsymbol{H}(\boldsymbol{\theta}; z) - \mathbb{E}_{p_{\boldsymbol{\theta}}(z)}(\boldsymbol{H}(\boldsymbol{\theta}; z))\|_{sp}^2\right) \leq \sigma_2^2$ we can directly obtain that

$$\frac{1}{K}\sum_{k=1}^{K} \mathrm{Var}(\boldsymbol{H}(\boldsymbol{\theta}^m, z^k)) \leq \sigma_2^2.$$

Then, for $\boldsymbol{B}^m$, by letting $t = b_2\sqrt{\rho\epsilon}$ in the Bernstein's inequality in Lemma B.4, we should have

$$P\big(\|\boldsymbol{B}^m - \nabla^2 \mathcal{J}(\boldsymbol{\theta}^m)\|_{\mathrm{sp}} \geq b_2\sqrt{\rho\epsilon}\big) \leq 2d^2 \exp\left(-\frac{Kb_2^2\rho\epsilon/2}{1/K\sum_{k=1}^K \mathrm{Var}(\boldsymbol{H}(\boldsymbol{\theta}^m, z_k)) + b_2 M_2\sqrt{\rho\epsilon}/3}\right).$$

Next, we have the following bound for R.H.S of the above inequality:

$$\frac{Kb_2^2\rho\epsilon/2}{1/K\sum_{k=1}^K \mathrm{Var}(\boldsymbol{H}(\boldsymbol{\theta}^m, z_k)) + b_2 M_2\sqrt{\rho\epsilon}/3} \geq \frac{Kb_2^2\rho\epsilon/2}{\sigma_2^2 + b_2 M_2\sqrt{\rho\epsilon}/3} = \frac{K}{4}\frac{2}{\sigma_2^2/b_2^2\rho\epsilon + M_2/3b_2\sqrt{\rho\epsilon}}.$$

Note that the last term has the harmonic mean of $b_2^2\rho\epsilon/\sigma_2^2$ and $3b_2\sqrt{\rho\epsilon}/M_2$, then by the property of harmonic means, we should have:

$$\frac{2}{\sigma_2^2/b_2^2\rho\epsilon + M_2/3b_2\sqrt{\rho\epsilon}} \geq \min\left\{b_2^2\rho\epsilon/\sigma_2^2, 3b_2\sqrt{\rho\epsilon}/M_2\right\},$$

and this will give us

$$P\big(\|\boldsymbol{B}^m - \nabla^2 \mathcal{J}(\boldsymbol{\theta}^m)\|_{\mathrm{sp}} \geq b_2\sqrt{\rho\epsilon}\big) \leq 2d^2 \exp\left(-\frac{K}{4}\min\{b_2^2\rho\epsilon/\sigma_2^2, 3b_2\sqrt{\rho\epsilon}/M_2\}\right).$$

Then we let

$$\delta_2 = 2d^2 \exp\left(-\frac{K}{4}\min\{b_2^2\rho\epsilon/\sigma_2^2, 3b_2\sqrt{\rho\epsilon}/M_2\}\right),$$

$$\text{i.e. } K = 4\max\{M_2/3b_2\sqrt{\rho\epsilon}, \sigma_2^2/b_2^2\rho\epsilon\}\log(2d^2/\delta_2) = \mathcal{O}\big(1/\epsilon^2\log(1/\delta_2)\big)$$

we can obtain $P\big(\|\boldsymbol{B}^m - \mathbb{E}_{p_{\boldsymbol{\theta}}(\boldsymbol{z})}(\boldsymbol{B}^m)\|_{\mathrm{sp}} \geq b_2\sqrt{\rho\epsilon}\big) \leq \delta_2$, which means $\|\boldsymbol{B}^m - \mathbb{E}_{p_{\boldsymbol{\theta}}(\boldsymbol{z})}(\boldsymbol{B}^m)\|_{\mathrm{sp}} \leq b_2\sqrt{\rho\epsilon}$ hold with probability at least $1 - \delta_2$.

Finally, letting $c_2 = \frac{1}{72} + b_2$, we can obtain that

$$\|\boldsymbol{B}^m - \nabla \mathcal{J}(\boldsymbol{\theta}^m)\|_{\mathrm{sp}} \leq \|\mathbb{E}_{p_{\boldsymbol{\theta}}(\boldsymbol{z})}(\boldsymbol{B}^m) - \nabla \mathcal{J}(\boldsymbol{\theta}^m)\|_{\mathrm{sp}} + \|\boldsymbol{B}^m - \mathbb{E}_{p_{\boldsymbol{\theta}}(\boldsymbol{z})}(\boldsymbol{B}^m)\|_{\mathrm{sp}}$$

$$\leq (\frac{1}{72} + b_2)\sqrt{\rho\epsilon} = c_2\sqrt{\rho\epsilon}$$

holds with probability at least $1 - \delta_2$, which concludes our proof.

$\square$

**Lemma B.6.** *Denote $\boldsymbol{\Delta}^m = \arg\min_{\boldsymbol{\Delta}} \tilde{c}_m(\boldsymbol{\Delta})$, then we will have:*

$$\boldsymbol{g}^m + \boldsymbol{B}^m\boldsymbol{\Delta}^m + \frac{\rho}{2}\|\boldsymbol{\Delta}^m\|_2 \boldsymbol{\Delta}^m = 0, \tag{8}$$

$$\boldsymbol{B}^m + \frac{\rho}{2}\|\boldsymbol{\Delta}^m\|_2 I \succeq 0, \tag{9}$$

$$\tilde{c}_m(\boldsymbol{\Delta}^m) \leq -\frac{\rho}{12}\|\boldsymbol{\Delta}^m\|_2^3. \tag{10}$$

*where $\boldsymbol{P} \succeq \boldsymbol{Q}$ indicates that $\boldsymbol{P} - \boldsymbol{Q}$ is a positive semidefinite matrix, i.e. the metric $\boldsymbol{B}^m + \frac{\rho}{2}\|\boldsymbol{\Delta}^m\|_2 I$ is always positive semidefinite.*

*Proof.* It is easy to verify that $\tilde{c}_m(\boldsymbol{\Delta})$, which is defined in (5), is second-order differentiable. Since $\boldsymbol{\Delta}^m$ is the global optimal, we must have $\nabla_{\boldsymbol{\Delta}^m}\tilde{c}_m(\boldsymbol{\Delta}^m) = 0$ and $\nabla^2_{\boldsymbol{\Delta}^m}\tilde{c}_m(\boldsymbol{\Delta}^m) \succeq 0$, i.e.

$$\nabla_{\boldsymbol{\Delta}^m}\tilde{c}_m(\boldsymbol{\Delta}^m) = \boldsymbol{g}^m + \boldsymbol{B}^m\boldsymbol{\Delta}^m + \frac{\rho}{2}\|\boldsymbol{\Delta}^m\|_2 \boldsymbol{\Delta}^m = 0,$$

$$\nabla^2_{\boldsymbol{\Delta}^m}\tilde{c}_m(\boldsymbol{\Delta}^m) = \boldsymbol{B}^m + \frac{\rho}{2}\|\boldsymbol{\Delta}^m\|_2 I \succeq 0.$$

These two equations are exactly (8) and (9). Now for (8), by computing the dot product of L.H.S. and $\boldsymbol{\Delta}^m$, we will have:

$$(\boldsymbol{\Delta}^m)^\top \boldsymbol{g}^m + (\boldsymbol{\Delta}^m)^\top \boldsymbol{B}^m\boldsymbol{\Delta}^m + \frac{\rho}{2}\|\boldsymbol{\Delta}^m\|_2^3 = 0. \tag{11}$$

And for (9), similarly by multiplying $(\mathbf{\Delta}^m)^\top$ and $\mathbf{\Delta}^m$ in both sides, we will have:

$$(\mathbf{\Delta}^m)^\top \mathbf{B}^m \mathbf{\Delta}^m + \frac{\rho}{2}\|\mathbf{\Delta}^m\|_2^3 \geq 0 \quad \text{and} \quad -(\mathbf{\Delta}^m)^\top \mathbf{B}^m \mathbf{\Delta}^m \leq \frac{\rho}{2}\|\mathbf{\Delta}^m\|_2^3. \tag{12}$$

Then, for $\tilde{c}_m(\mathbf{\Delta}^m)$, from (11) we will have:

$$\tilde{c}_m(\mathbf{\Delta}^m) = (\mathbf{\Delta}^m)^\top \mathbf{g}^m + \frac{1}{2}(\mathbf{\Delta}^m)^\top \mathbf{B}^m \mathbf{\Delta}^m + \frac{\rho}{6}\|\mathbf{\Delta}^m\|_2^3,$$

$$= (\mathbf{\Delta}^m)^\top \mathbf{g}^m + (\mathbf{\Delta}^m)^\top \mathbf{B}^m \mathbf{\Delta}^m + \frac{\rho}{2}\|\mathbf{\Delta}^m\|_2^3 - \frac{1}{2}(\mathbf{\Delta}^m)^\top \mathbf{B}^m \mathbf{\Delta}^m - \frac{\rho}{3}\|\mathbf{\Delta}^m\|_2^3,$$

$$= -\frac{1}{2}(\mathbf{\Delta}^m)^\top \mathbf{B}^m \mathbf{\Delta}^m - \frac{\rho}{3}\|\mathbf{\Delta}^m\|_2^3.$$

Now by (12), we will have:

$$\tilde{c}_m(\mathbf{\Delta}^m) = -\frac{1}{2}(\mathbf{\Delta}^m)^\top \mathbf{B}\mathbf{\Delta}^m - \frac{\rho}{3}\|\mathbf{\Delta}^m\|_2^3 \leq \frac{\rho}{4}\|\mathbf{\Delta}^m\|_2^3 - \frac{\rho}{3}\|\mathbf{\Delta}^m\|_2^3 = -\frac{\rho}{12}\|\mathbf{\Delta}^m\|_2^3.$$

which is exactly (10). $\qquad\square$

**Lemma B.7.** *If $0 < c_1 \leq \frac{3}{8}$ in Lemma B.3 and $0 < c_2 \leq \frac{1}{4}$ in Lemma B.5, then we will have:*

$$\|\mathbf{\Delta}^m\|_2 \geq \sqrt{\frac{1}{2\rho}\big(\|\nabla_{\boldsymbol{\theta}^{m+1}}\mathcal{J}(\boldsymbol{\theta}^{m+1})\|_2 - \frac{1}{2}\epsilon\big)}. \tag{13}$$

*Proof.* For (13), we first have:

$$\|\nabla_{\boldsymbol{\theta}^{m+1}}\mathcal{J}(\boldsymbol{\theta}^{m+1})\|_2 = \|\nabla_{\boldsymbol{\theta}^{m+1}}\mathcal{J}(\boldsymbol{\theta}^{m+1}) - \nabla\mathcal{J}(\boldsymbol{\theta}^m) - \nabla^2\mathcal{J}(\boldsymbol{\theta}^m)\mathbf{\Delta}^m + \nabla\mathcal{J}(\boldsymbol{\theta}^m) + \nabla^2\mathcal{J}(\boldsymbol{\theta}^m)\mathbf{\Delta}^m\|_2,$$

$$\leq \|\nabla_{\boldsymbol{\theta}^{m+1}}\mathcal{J}(\boldsymbol{\theta}^{m+1}) - \nabla\mathcal{J}(\boldsymbol{\theta}^m) - \nabla^2\mathcal{J}(\boldsymbol{\theta}^m)\mathbf{\Delta}^m\|_2 + \|\nabla\mathcal{J}(\boldsymbol{\theta}^m) + \nabla^2\mathcal{J}(\boldsymbol{\theta}^m)\mathbf{\Delta}^m\|_2. \tag{14}$$

The first term in (14) can be bounded in (6) in Lemma B.1. We note that $\boldsymbol{\theta}^{m+1} - \boldsymbol{\theta}^m = \mathbf{\Delta}^m$, thus we have:

$$\|\nabla_{\boldsymbol{\theta}^{m+1}}\mathcal{J}(\boldsymbol{\theta}^{m+1}) - \nabla\mathcal{J}(\boldsymbol{\theta}^m) - \nabla^2\mathcal{J}(\boldsymbol{\theta}^m)\mathbf{\Delta}^m\|_2 \leq \frac{\rho}{2}\|\mathbf{\Delta}^m\|_2^2. \tag{15}$$

For the second term in (14), we have:

$$\|\nabla\mathcal{J}(\boldsymbol{\theta}^m) + \nabla^2\mathcal{J}(\boldsymbol{\theta}^m)\mathbf{\Delta}^m\|_2 = \|\nabla\mathcal{J}(\boldsymbol{\theta}^m) - \mathbf{g}^m + \mathbf{g}^m + (\nabla^2\mathcal{J}(\boldsymbol{\theta}^m) - \mathbf{B}^m)\mathbf{\Delta}^m + \mathbf{B}^m\mathbf{\Delta}^m\|_2,$$

$$\leq \|\nabla\mathcal{J}(\boldsymbol{\theta}^m) - \mathbf{g}^m\|_2 + \|(\nabla^2\mathcal{J}(\boldsymbol{\theta}^m) - \mathbf{B}^m)\mathbf{\Delta}^m\|_2 + \|\mathbf{g}^m + \mathbf{B}^m\mathbf{\Delta}^m\|_2. \tag{16}$$

Specifically, for terms in (16), we have following conditions.

- First term in (16). From Lemma B.3, we will have:

$$\|\nabla\mathcal{J}(\boldsymbol{\theta}^m) - \mathbf{g}^m\|_2 \leq c_1\epsilon. \tag{17}$$

- Second term in (16). From Lemma B.5, we will have:

$$\|(\nabla^2\mathcal{J}(\boldsymbol{\theta}^m) - \mathbf{B}^m)\mathbf{\Delta}^m\|_2 \leq c_2\sqrt{\rho\epsilon}\|\mathbf{\Delta}^m\|_2. \tag{18}$$

- Last term in (16). From (8) in Lemma B.6, we will have:

$$\mathbf{g}^m + \mathbf{B}^m\mathbf{\Delta}^m = -\frac{\rho}{2}\|\mathbf{\Delta}^m\|_2\mathbf{\Delta}^m \quad \rightarrow \quad \|\mathbf{g}^m + \mathbf{B}^m\mathbf{\Delta}^m\|_2 = \frac{\rho}{2}\|\mathbf{\Delta}^m\|_2^2. \tag{19}$$

Combining (15) to (19), for (14), we will have

$$\big\|\nabla_{\boldsymbol{\theta}^{m+1}}\mathcal{J}(\boldsymbol{\theta}^{m+1})\big\|_2 \leq \frac{\rho}{2}\|\mathbf{\Delta}^m\|_2^2 + c_1\epsilon + c_2\sqrt{\rho\epsilon}\|\mathbf{\Delta}^m\|_2 + \frac{\rho}{2}\|\mathbf{\Delta}^m\|_2^2,$$

$$= \rho\|\mathbf{\Delta}^m\|_2^2 + c_1\epsilon + c_2\sqrt{\rho\epsilon}\|\mathbf{\Delta}^m\|_2.$$

Note that

$$\sqrt{\rho\epsilon}\|\mathbf{\Delta}^m\|_2 = \sqrt{\rho\|\mathbf{\Delta}^m\|_2^2 \cdot \epsilon} \leq \frac{\rho\|\mathbf{\Delta}^m\|_2^2 + \epsilon}{2},$$

therefore we will have:

$$\|\nabla_{\boldsymbol{\theta}^{m+1}}\mathcal{J}(\boldsymbol{\theta}^{m+1})\|_2 \leq \rho\|\boldsymbol{\Delta}^m\|_2^2 + c_1\epsilon + \frac{c_2}{2}(\rho\|\boldsymbol{\Delta}^m\|_2^2 + \epsilon),$$

which implies that:

$$\|\boldsymbol{\Delta}^m\|_2^2 \geq \frac{1}{\rho(1 + \frac{c_2}{2})}(\|\nabla_{\boldsymbol{\theta}^{m+1}}\mathcal{J}(\boldsymbol{\theta}^{m+1})\|_2 - (c_1 + \frac{c_2}{2})\epsilon).$$

Now from $0 < c_1 \leq \frac{3}{8}$ and $0 < c_2 \leq \frac{1}{4}$, we can easily prove that $1 + \frac{c_2}{2} \leq 2$ and $c_1 + \frac{c_2}{2} \leq \frac{1}{2}$, thus we will have

$$\|\boldsymbol{\Delta}^m\|_2^2 \geq \frac{1}{2\rho}(\|\nabla_{\boldsymbol{\theta}^{m+1}}\mathcal{J}(\boldsymbol{\theta}^{m+1})\|_2 - \frac{1}{2}\epsilon),$$

which is exactly (13). $\qquad\square$

**Lemma B.8.** *If $0 < c_2 \leq \frac{1}{4}$ in Lemma B.5, then for $\boldsymbol{\Delta}^m = \arg\min_{\boldsymbol{\Delta}} \tilde{c}_m(\boldsymbol{\Delta})$ we will have:*

$$\|\boldsymbol{\Delta}^m\|_2 \geq -\frac{2}{3\rho}\lambda_{\min}(\nabla_{\boldsymbol{\theta}^{m+1}}^2\mathcal{J}(\boldsymbol{\theta}^{m+1})) - \frac{1}{6}\sqrt{\frac{\epsilon}{\rho}}, \tag{20}$$

*where $\lambda_{\min}(\boldsymbol{A})$ denotes the minimum eigenvalue of a square matrix $\boldsymbol{A}$.*

*Proof.* Since the Hessian for $\mathcal{J}(\boldsymbol{\theta})$ is Lipschitz continuous as in Assumption 3.6 (i), we will have:

$$\|\nabla^2\mathcal{J}(\boldsymbol{\theta}^m + \boldsymbol{\Delta}^m) - \nabla^2\mathcal{J}(\boldsymbol{\theta}^m)\|_{\text{sp}} \leq \rho\|\boldsymbol{\Delta}^m\|_2,$$

which implies that

$$\nabla^2\mathcal{J}(\boldsymbol{\theta}^m + \boldsymbol{\Delta}^m) - \nabla^2\mathcal{J}(\boldsymbol{\theta}^m) \succeq -\rho\|\boldsymbol{\Delta}^m\|I. \tag{21}$$

From Lemma B.5, we have

$$\|\boldsymbol{B}^m - \nabla^2\mathcal{J}(\boldsymbol{\theta}^m)\|_{\text{sp}} \leq c_2\sqrt{\rho\epsilon},$$

which also implies that

$$\nabla^2\mathcal{J}(\boldsymbol{\theta}^m) - \boldsymbol{B} \succeq -c_2\sqrt{\rho\epsilon}I, \tag{22}$$

Finally, from (9), we will have

$$\boldsymbol{B} \succeq -\frac{\rho}{2}\|\boldsymbol{\Delta}^m\|_2I. \tag{23}$$

Combining (21) to (23) together, we will have:

$$\nabla^2\mathcal{J}(\boldsymbol{\theta}^m + \boldsymbol{\Delta}^m) = \nabla^2\mathcal{J}(\boldsymbol{\theta}^m + \boldsymbol{\Delta}^m) - \nabla^2\mathcal{J}(\boldsymbol{\theta}^m) + \nabla^2\mathcal{J}(\boldsymbol{\theta}^m) - \boldsymbol{B} + \boldsymbol{B},$$

$$\succeq -\rho\|\boldsymbol{\Delta}^m\|_2I - c_2\sqrt{\rho\epsilon}I - \frac{\rho}{2}\|\boldsymbol{\Delta}^m\|_2I = -(c_2\sqrt{\rho\epsilon} + \frac{3}{2}\rho\|\boldsymbol{\Delta}^m\|_2)I.$$

Note that $\boldsymbol{\theta}^{m+1} = \boldsymbol{\theta}^m + \boldsymbol{\Delta}^m$, and from the definition of $\lambda_{\min}(\cdot)$, we will have:

$$\lambda_{\min}(\nabla_{\boldsymbol{\theta}^{m+1}}^2\mathcal{J}(\boldsymbol{\theta}^{m+1})) \geq -\left(c_2\sqrt{\rho\epsilon} + \frac{3}{2}\rho\|\boldsymbol{\Delta}^m\|_2\right),$$

which can deduce that:

$$\|\boldsymbol{\Delta}^m\|_2 \geq -\frac{2}{3\rho}\lambda_{\min}\left(\nabla_{\boldsymbol{\theta}^{m+1}}^2\mathcal{J}(\boldsymbol{\theta}^{m+1})\right) - \frac{2c_2}{3}\sqrt{\frac{\epsilon}{\rho}}.$$

Since $0 < c_2 \leq \frac{1}{4}$, we will have

$$\|\boldsymbol{\Delta}^m\|_2 \geq -\frac{2}{3\rho}\lambda_{\min}(\nabla_{\boldsymbol{\theta}^{m+1}}^2\mathcal{J}(\boldsymbol{\theta}^{m+1})) - \frac{1}{6}\sqrt{\frac{\epsilon}{\rho}},$$

which is exactly (20). $\qquad\square$

**Lemma B.9.** *If $0 < c_1 < \frac{1}{32}$ in Lemma B.3 and $0 < c_2 < \frac{1}{36}$ in Lemma B.5, then there exists a constant $C = \frac{1}{96} - (\frac{c_1}{2} + \frac{c_2}{8}) > 0$ such that $\mathcal{J}(\boldsymbol{\theta}^{m+1}) - \mathcal{J}(\boldsymbol{\theta}^m) \leq -C\sqrt{\frac{\epsilon^3}{\rho}}$ if we have $\tilde{c}_m(\boldsymbol{\Delta}^m) \leq -\frac{1}{96}\sqrt{\frac{\epsilon^3}{\rho}}$ for $\boldsymbol{\Delta}^m = \arg\min_{\boldsymbol{\Delta}} \tilde{c}_m(\boldsymbol{\Delta})$.*

*Proof.* Recall that (7) in Lemma B.1 gives us

$$\mathcal{J}(\boldsymbol{\theta}^m + \boldsymbol{\Delta}^m) - \mathcal{J}(\boldsymbol{\theta}^m) \leq (\nabla\mathcal{J}(\boldsymbol{\theta}^m))^\top\boldsymbol{\Delta}^m + \frac{1}{2}(\boldsymbol{\Delta}^m)^\top\nabla^2\mathcal{J}(\boldsymbol{\theta}^m)\boldsymbol{\Delta}^m + \frac{\rho}{6}\|\boldsymbol{\Delta}^m\|_2^3.$$

And by the definition of $\tilde{c}_m(\cdot)$, we have:

$$\tilde{c}_m(\boldsymbol{\Delta}) = (\boldsymbol{g}^m)^\top\boldsymbol{\Delta} + \frac{1}{2}\boldsymbol{\Delta}^\top\boldsymbol{B}^m\boldsymbol{\Delta} + \frac{\rho}{6}\|\boldsymbol{\Delta}\|_2^3.$$

Therefore we can obtain that:

$$\mathcal{J}(\boldsymbol{\theta}^m + \boldsymbol{\Delta}^m) - \mathcal{J}(\boldsymbol{\theta}^m) = \tilde{c}_m(\boldsymbol{\theta}^m) + (\nabla\mathcal{J}(\boldsymbol{\theta}^m) - \boldsymbol{g})^\top\boldsymbol{\Delta}^m + \frac{1}{2}(\boldsymbol{\Delta}^m)^\top(\nabla^2\mathcal{J}(\boldsymbol{\theta}^m) - \boldsymbol{B})\boldsymbol{\Delta}^m.$$

From Lemma B.3, we will have:

$$(\nabla\mathcal{J}(\boldsymbol{\theta}^m) - \boldsymbol{g})^\top\boldsymbol{\Delta}^m \leq c_1\epsilon\|\boldsymbol{\Delta}^m\|_2.$$

And from Lemma B.5, we will have:

$$\frac{1}{2}(\boldsymbol{\Delta}^m)^\top(\nabla^2\mathcal{J}(\boldsymbol{\theta}^m) - \boldsymbol{B})\boldsymbol{\Delta}^m \leq \frac{c_2}{2}\sqrt{\rho\epsilon}\|\boldsymbol{\Delta}^m\|_2^2.$$

Combining these two parts will give us:

$$\mathcal{J}(\boldsymbol{\theta}^m + \boldsymbol{\Delta}^m) - \mathcal{J}(\boldsymbol{\theta}^m) \leq \tilde{c}_m(\boldsymbol{\Delta}^m) + c_1\epsilon\|\boldsymbol{\Delta}^m\|_2 + \frac{c_2}{2}\sqrt{\rho\epsilon}\|\boldsymbol{\Delta}^m\|_2^2.$$

Now we prove separately under two cases:

**Case 1:** $\|\boldsymbol{\Delta}^m\|_2 > \frac{1}{2}\sqrt{\frac{\epsilon}{\rho}}$. In such case, we have

$$c_1\epsilon\|\boldsymbol{\Delta}^m\|_2 + \frac{c_2}{2}\sqrt{\rho\epsilon}\|\boldsymbol{\Delta}^m\|_2^2 = \left(\frac{c_1\epsilon}{\|\boldsymbol{\Delta}^m\|_2^2} + \frac{c_2\sqrt{\rho\epsilon}}{2\|\boldsymbol{\Delta}^m\|_2}\right)\|\boldsymbol{\Delta}^m\|_2^3,$$

$$\leq (4c_1 + c_2)\rho\|\boldsymbol{\Delta}^m\|_2^3.$$

Then for $\tilde{c}_m(\boldsymbol{\Delta}^m)$, recall that we have $\tilde{c}_m(\boldsymbol{\Delta}^m) \leq -\frac{1}{12}\rho\|\boldsymbol{\Delta}^m\|_2^3$ in (10), and combining these two together will give us:

$$\mathcal{J}(\boldsymbol{\theta}^{m+1}) - \mathcal{J}(\boldsymbol{\theta}^m) \leq \tilde{c}_m(\boldsymbol{\Delta}^m) + c_1\epsilon\|\boldsymbol{\Delta}^m\|_2 + \frac{c_2}{2}\sqrt{\rho\epsilon}\|\boldsymbol{\Delta}^m\|_2^2$$

$$\leq -\left(\frac{1}{12} - (4c_1 + c_2)\right)\rho\|\boldsymbol{\Delta}^m\|_2^3$$

Finally from $0 < c_1 < \frac{1}{32}$ and $0 < c_2 < \frac{1}{36}$, we can prove that $\frac{1}{12} - (4c_1 + c_2) > 0$ always holds. Therefore, with $\|\boldsymbol{\Delta}^m\|_2 \geq \frac{1}{2}\sqrt{\frac{\epsilon}{\rho}}$, we now have:

$$\mathcal{J}(\boldsymbol{\theta}^m + \boldsymbol{\Delta}^m) - \mathcal{J}(\boldsymbol{\theta}^m) \leq -\left(\frac{1}{12} - (4c_1 + c_2)\right)\rho \cdot \frac{1}{8}\sqrt{\frac{\epsilon^3}{\rho^3}} = -\left(\frac{1}{96} - \left(\frac{c_1}{2} + \frac{c_2}{8}\right)\right)\sqrt{\frac{\epsilon^3}{\rho}}. \quad (24)$$

**Case 2:** $\|\boldsymbol{\Delta}^m\|_2 \leq \frac{1}{2}\sqrt{\frac{\epsilon}{\rho}}$. In such case, we have

$$c_1\epsilon\|\boldsymbol{\Delta}^m\|_2 + \frac{c_2}{2}\sqrt{\rho\epsilon}\|\boldsymbol{\Delta}^m\|_2^2 \leq c_1\epsilon \cdot \frac{1}{2}\sqrt{\frac{\epsilon}{\rho}} + \frac{c_2}{2}\sqrt{\rho\epsilon} \cdot \frac{1}{4}\frac{\epsilon}{\rho} = \left(\frac{c_1}{2} + \frac{c_2}{8}\right)\sqrt{\frac{\epsilon^3}{\rho}}.$$

And we also have $\tilde{c}_m(\boldsymbol{\Delta}^m) \leq -\frac{1}{96}\sqrt{\frac{\epsilon^3}{\rho}}$ from the assumption in Lemma B.9. Then combining these two bounds will give us:

$$\mathcal{J}(\boldsymbol{\theta}^{m+1}) - \mathcal{J}(\boldsymbol{\theta}^m) \leq \tilde{c}_m(\boldsymbol{\Delta}^m) + c_1\epsilon\|\boldsymbol{\Delta}^m\|_2 + \frac{c_2}{2}\sqrt{\rho\epsilon}\|\boldsymbol{\Delta}^m\|_2^2$$

$$\leq -\left(\frac{1}{96} - \left(\frac{c_1}{2} + \frac{c_2}{8}\right)\right)\sqrt{\frac{\epsilon^3}{\rho}} \tag{25}$$

Combining (24) and (25) together, we prove that $\mathcal{J}(\boldsymbol{\theta}^{m+1}) - \mathcal{J}(\boldsymbol{\theta}^m) \leq -C\sqrt{\frac{\epsilon^3}{\rho}}$ for $C = \frac{1}{96} - \left(\frac{c_1}{2} + \frac{c_2}{8}\right) > 0$. $\qquad\square$

**Lemma B.10.** $\exists M \leq \frac{\sqrt{\rho}(\mathcal{J}(\boldsymbol{\theta}^1) - \mathcal{J}(\boldsymbol{\theta}^*))}{C\epsilon^{1.5}} + 1$, such that $\mathcal{J}(\boldsymbol{\theta}^{M+1}) - \mathcal{J}(\boldsymbol{\theta}^M) \geq -C\sqrt{\frac{\epsilon^3}{\rho}}$ where $C$ is the constant given in Lemma B.9

*Proof.* We will prove this lemma by contradiction. Suppose that:

$$\forall m \leq \frac{\sqrt{\rho}(\mathcal{J}(\boldsymbol{\theta}^1) - \mathcal{J}(\boldsymbol{\theta}^*))}{C\epsilon^{1.5}} + 1, \quad \mathcal{J}(\boldsymbol{\theta}^{m+1}) - \mathcal{J}(\boldsymbol{\theta}^m) \leq -C\sqrt{\frac{\epsilon^3}{\rho}}$$

And obviously, we should have:

$$\sum_{m=1}^{M} \mathcal{J}(\boldsymbol{\theta}^m) - \mathcal{J}(\boldsymbol{\theta}^{m+1}) = \mathcal{J}(\boldsymbol{\theta}^1) - \mathcal{J}(\boldsymbol{\theta}^{M+1}) \leq \mathcal{J}(\boldsymbol{\theta}^1) - \mathcal{J}(\boldsymbol{\theta}^*) \quad , \quad \forall M \geq 1$$

Thus, we let $M_0 = \frac{\sqrt{\rho}(\mathcal{J}(\boldsymbol{\theta}^1) - \mathcal{J}(\boldsymbol{\theta}^*))}{K\epsilon^{1.5}} + 1$, and we will have that:

$$\sum_{m=1}^{M_0} \mathcal{J}(\boldsymbol{\theta}^m) - \mathcal{J}(\boldsymbol{\theta}^{m+1}) \geq \left(\frac{\sqrt{\rho}(\mathcal{J}(\boldsymbol{\theta}^1) - \mathcal{J}(\boldsymbol{\theta}^*))}{K\epsilon^{1.5}} + 1\right)K\sqrt{\frac{\epsilon^3}{\rho}}$$

$$= \mathcal{J}(\boldsymbol{\theta}^1) - \mathcal{J}(\boldsymbol{\theta}^*) + K\sqrt{\frac{\epsilon^3}{\rho}} > \mathcal{J}(\boldsymbol{\theta}^1) - \mathcal{J}(\boldsymbol{\theta}^*)$$

which contradicts with previous condition. Thus we prove the original lemma. $\qquad\square$

### B.2 Proof of Theorem 3.7

*Proof.* From Lemma B.9 and Lemma B.10, we will have that $\tilde{c}_m(\boldsymbol{\Delta}^M) \geq -\frac{1}{96}\sqrt{\frac{\epsilon^3}{\rho}}$ for a

$$M \leq \frac{\sqrt{\rho}(\mathcal{J}(\boldsymbol{\theta}^1) - \mathcal{J}(\boldsymbol{\theta}^*))}{C\epsilon^{1.5}} + 1,$$

where $C = \frac{1}{96} - \left(\frac{c_1}{2} + \frac{c_2}{8}\right)$. Then, (10) in Lemma B.6 implies that:

$$-\frac{1}{12}\rho\|\boldsymbol{\Delta}^m\|_2^3 \geq \tilde{c}_m(\boldsymbol{\Delta}^M) \geq -\frac{1}{96}\sqrt{\frac{\epsilon^3}{\rho}} \quad \rightarrow \quad \|\boldsymbol{\Delta}^m\|_2 \leq \frac{1}{2}\sqrt{\frac{\epsilon}{\rho}}.$$

Then from Lemma B.7, we will have

$$\sqrt{\frac{1}{2\rho}(\|\nabla_{\boldsymbol{\theta}^{M+1}}\mathcal{J}(\boldsymbol{\theta}^{M+1})\|_2 - \frac{1}{2}\epsilon)} \leq \|\boldsymbol{\Delta}^m\|_2 \leq \frac{1}{2}\sqrt{\frac{\epsilon}{\rho}} \quad \rightarrow \quad \|\nabla_{\boldsymbol{\theta}^{M+1}}\mathcal{J}(\boldsymbol{\theta}^{M+1})\|_2 \leq \epsilon.$$

From Lemma B.8, we will have

$$-\frac{2}{3\rho}\lambda_{\min}(\nabla_{\boldsymbol{\theta}^{M+1}}^2\mathcal{J}(\boldsymbol{\theta}^{M+1})) - \frac{1}{6}\sqrt{\frac{\epsilon}{\rho}} \leq \|\boldsymbol{\Delta}^m\|_2 \leq \frac{1}{2}\sqrt{\frac{\epsilon}{\rho}} \rightarrow \lambda_{\min}(\nabla_{\boldsymbol{\theta}^{M+1}}^2\mathcal{J}(\boldsymbol{\theta}^{M+1})) \geq -\sqrt{\rho\epsilon}.$$

Therefore, we have proved that $\boldsymbol{\theta}^{M+1}$ is a second-order stationary point defined in Definition 3.4. $\quad\square$

# C    Comparison among Different Hyper-parameter Optimization Methods

Table 3 shows the comparison among different optimization methods. The representative derivative-free methods are in the first two rows. As introduced in related works (Section 2.1), they do not have any convergence guarantee. Methods utilizing hyper-gradient based on stochastic relaxation are in the middle 5 rows. Although all these methods have first-order convergence guarantee, only our proposed method with cubic regularization is able to converge to second-order stationary point. Furthermore, only our proposed method considers inexact lower-level objective. The two methods at bottom are designed for stochastic bi-level optimization. Although they consider inexact lower-level objective, they require upper-level objective to be differentiable, which means that they cannot be applied in hyper-parameter optimization problem considered in this paper, where hyper-gradients are unavailable.

Table 3: Comparison of different hyper-parameter optimization algorithms.

| Method | Require upper-level differentiable | Stochastic relaxation | Inexact lower-level | Convergence 1st-order | 2nd-order |
|---|---|---|---|---|---|
| Random search [2] | ✗ | ✗ | ✗ | ✗ | ✗ |
| Bayesian optimization [11] | ✗ | ✗ | ✗ | ✗ | ✗ |
| Gradient descent [14] | ✗ | ✓ | ✗ | ✓ | ✗ |
| Natural gradient [35] | ✗ | ✓ | ✗ | ✓ | ✗ |
| Newton's method [30] | ✗ | ✓ | ✗ | ✓ | ✗ |
| Cubic regularization | ✗ | ✓ | ✗ | ✓ | ✓ |
| Cubic (inexact) | ✗ | ✓ | ✓ | ✓ | ✓ |
| StocBiO [7] | ✓ | ✗ | ✓ | ✓ | ✗ |
| SUSTAIN [41] | ✓ | ✗ | ✓ | ✓ | ✗ |
| iNEON [31] | ✓ | ✗ | ✓ | ✓ | ✓ |

# D  Mathematical Formulation of Problems in the Experiments

For both experiments on synthetic data and hyper-parameter optimization for knowledge graph completion, their original formulations before stochastic relaxtaion are given by:

$$z^* = \arg\min_z H(w^*(z), z; D_{\text{val}}) \text{ s.t. } w^*(z) = \arg\min_w F(w, z; D_{\text{tra}})$$

For experiment on synthetic data, its mathematical formulation under stochastic relaxtaion can be referred to Eq (26) (the same as Eq (1) in Section 3.1), where $z$ corresponds to feature mask, $p_\theta$ uses the sigmoid function and details can be found in Section 4.1. We use AUC as the validation evaluation metric.

$$\boldsymbol{\theta}^* = \arg\min_{\boldsymbol{\theta}} \left\{ \mathcal{J}(\boldsymbol{\theta}) = \mathbb{E}_{z \sim p_{\boldsymbol{\theta}}(z)}[H(w^*(z), z; D_{\text{val}})] \right\} \text{ s.t. } w^*(z) = \arg\min_w F(w, z; D_{\text{tra}}), \quad (26)$$

For hyper-parameter optimization for knowledge graph completion, the mathematical formulation can also be referred to Eq (26), where $z$ corresponds to a set of categorical hyper-parameters (details in Table 4), $p_\theta$ uses the softmax-like function as defined in Section 4.2.1. We use MRR as the validation evaluation metric.

For schedule search on learning with noisy training data, the mathematical formulation before stochastic relaxation can be written as:

$$z^* = \arg\min_z M_{\text{val}}(w^*(R_z)) \text{ s.t. } w^*(z) = \arg\min_w L_{\text{tra}}(w, R_z)$$

$R_z$ is the schedule function parameterized in Section 4.2.2. With stochastic relaxation, its formulation is changed to:

$$\begin{aligned} \boldsymbol{\theta}^* = \arg\min_{\boldsymbol{\theta}} \mathcal{J}(\boldsymbol{\theta}) \ \text{ s.t. } \ w^*(z) = \arg\min_w L_{\text{tra}}(w, R_z) \\ \text{where } \ \mathcal{J}(\boldsymbol{\theta}) = \mathbb{E}_{z \sim p_{\boldsymbol{\theta}}(z)}[M_{\text{val}}(w^*(R_z))] \end{aligned} \quad (27)$$

Here, $z$ is sampled from the corresponding distribution $p_\theta$. We use Beta distribution for each element of $\beta$ and use Dirichlet distribution for $\alpha$. The validation evaluation metric is accuracy.

# E  Implementation Details

## E.1  Experiments on Synthetic Data

For experiment on synthetic data, we use a linear model with single layer as feature classifier and use AdamW with default hyper-parameter settings as the optimizer. We train the model for 40 epochs to obtain the results in Section 4.1.

## E.2  Experiments for Hyper-parameter Optimization for Knowledge Graph Completion

In experiments for hyper-parameter optimization for knowledge graph completion, we use traditional embedding based model. We set batch size as 128 and dimension size as 100. Dropout rate is set as 0.2 and we choose Adam [42] as the optimizer. For different hyper-parameter settings, we train the model for 100000 iterations. The search space for other hyper-parameters is shown in Table 4.

Table 4: The hyper-parameter search space of knowledge graph completion experiments.

| Hyper-parameter | search space |
|---|---|
| negative sampling number | {32, 128, 256, 1024} |
| regularizer | {FRO, NUC, DURA, None} |
| loss function | {MR, BCE_(mean, sum, adv), CE} |
| gamma (MR) | {2, 4, 8} |
| initializer | {uniform, xavier_norm, xavier_uniform, normal} |
| learning rate | {0.01, 0.001} |
| score function | {TransE, RotatE, ComplEx} |

Details on some acronyms in Table 4 are introduced below.

**Regularizer.** To avoid overfitting of the embedding model, the regularization technique is considered, including Frobenius norm (FRO) [43], Nuclear norm (NUC) [44] and DURA [45].

**Scoring function.** As an important component for knowledge graph learning, the choice of scoring function influence a lot to the learning performance. In the experiment in this paper, we set TransE [46], RotatE [47] and ComplEx [48] as search space choices.

**Loss function.** We mainly choose three types of loss functions: margin ranking (MR) loss [46], binary cross entropy (BCE) loss with its variants BCE_mean, BCE_adv [49] and BCE_sum [48], and finally the cross entropy (CE) loss [44].

Statistics of different benchmark knowledge graph data sets is in Table 5.

Table 5: Statistics of datasets for hyper-parameter optimization for knowledge graph completion.

| dataset | #entity | #relation | #training | #validation | #test |
|---------|---------|-----------|-----------|-------------|-------|
| WN18RR | 41k | 11 | 87k | 3k | 3k |
| FB15k237 | 15k | 237 | 272k | 18k | 20k |

### E.3 Experiments for Schedule Search on Learning with Noisy Training Data

In experiments for schedule search on learning with noisy training data, we use a 5-layer CNN similar to LeNet as the model for image classification and use Adam as the optimizer. Each model is trained for 200 epochs.

Moreover, here we illustrate the noise setting in the experiments. The label transition matrices $M$ for symmetric/pair flipping noise on an example 5-way classification problem are illustrated in Figure 5. Here each $M_{ij} = \Pr(\tilde{y} = j | y = i)$ is the probability that the noisy label $\tilde{y}$ is $j$ given that the true label $y$ is $i$.

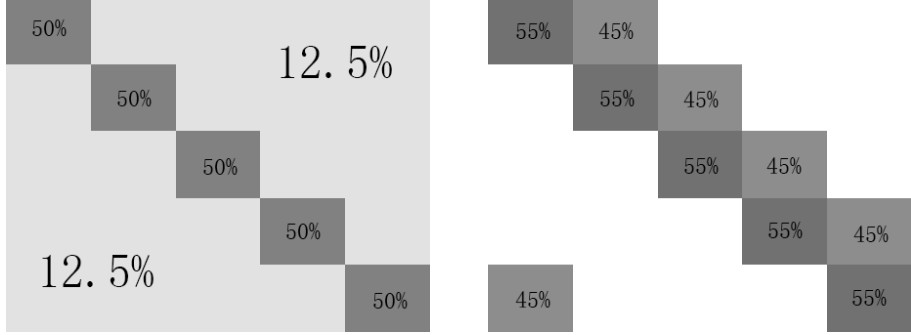

Figure 5: The two types of label noise used. Left: 50% symmetric flipping noise; right: 45% pair flipping noise.