# OpenReview forum: "Efficient Hyper-parameter Optimization with Cubic Regularization"
_NeurIPS.cc/2023/Conference — NeurIPS 2023 poster_

### Official Review · Reviewer_yqty · 2023-06-21

**Soundness:** 3 good
**Presentation:** 4 excellent
**Contribution:** 2 fair
**Rating:** 5
**Confidence:** 4

**Summary:**

{Summary}
This paper proposes a stochastic cubic regularization type algorithm for hyperparameter optimization that does not depend on hyper-gradients. Theoretical analysis shows that the proposed method can converge to approximate second order stationary points with lower sample complexity than that of first-order optimization methods, which can only find first-order stationary points. Experiments demonstrate the effectiveness of the proposed method using both synthetic and real data.

**Strengths:**

{Strengths}
1. This paper is clearly written and provides a new algorithm to solve nonconvex bilevel hyperparameter optimization problems (with stochastic relaxation).
2. In this setting, solving the cubic subproblem takes little time, as the cubic problem dimension is usually very small.
3. Theoretical analysis shows that the proposed cubic algorithm achieves lower sample complexity than that of first-order methods.



**Weaknesses:**

{Weakness and Questions}
1. The algorithm design is a direct application of cubic regularization, and therefore is not novel.
2. It is not clear what is the technical novelty in the convergence analysis compare to the analysis of inexact cubic regularization. The author mentions that the constructed $g$ and $B$ are not unbiased estimators of the gradient and Hessian. How does this challenge and affect the technical proof?
3. There are existing works on finding second-order stationary points of bi-level optimization, e.g., ``Efficiently Escaping Saddle Points in Bilevel Optimization''. Please discuss and compare with it.
4.  In the experiments, many curves are piece-wise flat (constant). Does that mean the hyperparameters do not change in those iterations?

**Questions:**

see the previous section.

**Limitations:**

see the previous section.

---

> ### Author Rebuttal · Authors · 2023-08-08
>
> Thank you for your comments on both the strength and weakness of our paper. For your questions raised in the weakness part, here are our answers to each of them:
>
> > Q1: The algorithm design is a direct application of cubic regularization, and therefore is not novel.
>
> A1: Our algorithm is different from existing works from the following perspectives:
> 1) in implementation, we need to compute gradient and Hessian from solving the lower-level problem, as in step 8 in Algorithm 1;
> 2) theory, we consider optimization with inexact gradient and Hessian, which is also different from directly using cubic regularization.
>
> > Q2: It is not clear what is the technical novelty in the convergence analysis compare to the analysis of inexact cubic regularization.
>
> A2: Thank you for raising this question. Our convergence analysis is different from existing analysis for inexact cubic regularization methods as the inexactness here comes from solving the lower-level problem. As such, we need to bound the error of estimated gradient and Hessian from the solution to lower-level problem, while existing analysis directly assumes error bound on estimating gradient or Hessian.
>
> > Q3: The author mentions that the constructed g and B are not unbiased estimators of the gradient and Hessian. How does this challenge and affect the technical proof?
>
> A3: As g (resp. B) are not unbiased estimators for gradient (resp. Hessian), the estimation error now contains two parts: stochastic noise and inexactness in solving lower-level problems. These parts need to be separately bounded in our proof. We will mention these differences in the final version of our paper.
>
> > Q4: There are existing works on finding second-order stationary points of bi-level optimization, e.g., ``Efficiently Escaping Saddle Points in Bilevel Optimization''. Please discuss and compare with it.
>
> A4: Thank you for pointing this out. We are aware of this paper and it is based on hyper-gradient for solving bi-level optimization problems. As such, it cannot be directly applied to our settings here as the hyper-gradient is unavailable due to discrete hyper-parameters or non-differentiable evaluation metrics (e.g., MRR in knowledge completion task, see Section 4.2.1). We will add some discussions on this paper in our final version.
>
> > Q5: In the experiments, many curves are piece-wise flat (constant). Does that mean the hyperparameters do not change in those iterations?
>
> A5: In our experiments, we report the best validation performances of found hyper-parameters. (similar figures can be referred to Figure 1 in [1] or Figure 2 in [2]) When curves keep piece-wise flat, it means that model with newly sampled hyper-parameter does not obtain better performance than the current optimum. We will add more descriptions to avoid any misunderstandings.
>
> [1] Initializing bayesian hyperparameter optimization via meta-learning, Proceedings of the AAAI Conference on Artificial Intelligence.
>
> [2] SMAC3: A versatile Bayesian optimization package for hyperparameter optimization, The Journal of Machine Learning Research.

---

> > ### Comment · Reviewer_yqty · 2023-08-21
> > **comment**
> >
> > I would like to thank the authors for their detailed response, which has addressed my technical quetions. After reading other reviewers' comments and authors' response, I decided to keep my current score.

---

> > > ### Author Response · Authors · 2023-08-21
> > > **Thanks!**
> > >
> > > We are pleased to know that our responses have addressed your previous questions. We are also willing to engage in more discussions if you have any other questions.

---

### Official Review · Reviewer_EwuD · 2023-07-04

**Soundness:** 3 good
**Presentation:** 2 fair
**Contribution:** 3 good
**Rating:** 6
**Confidence:** 3

**Summary:**

The authors propose a cubic regularization scheme for the outer loop optimization of bilevel optimization problems. The cubic regularization is very appealing since it can "avoid saddle points". Theoretically, authors extend the convergence results of the cubic regularization to inexact gradients to make it work for bilevel optimization. Empirically, authors show that their method achieves better performance on bilevel optimization tasks. Interestingly, they investigate the eigenvalues of the hessian.


**Strengths:**

The idea is very interesting and tackles the very important and hard problem of finding a proper outer procedure for bilevel optimization. Avoiding saddle points seems to be a very appealing property!

**Weaknesses:**

However, it feels like the writing of the paper could be polished (there are unmerged commit line 264 in the supplementary material.). A lot of statements are not properly backed. I am not sure to properly understand the interest of the proposed method.

**Questions:**


- "These methods [Bayesian optimization, Reinforcement learning] can easily get trapped in saddle points with poor performance" Could you provide references for this statement? I am not sure to understand why a Bayesian optimization algorithm would be stuck in a saddle point. From what I understand from the paper, hyper-gradient methods with vanilla first-order methods can be trapped in saddle point, and cubic regularization is here to help.
- What is the point of Lemma 3.1?
- Definition of what is called a saddle point could be recalled (i.e. negative eigenvalues of the hessian?)
- How does the proposed method compare to the use of l-BFGS once the hyper-gradient has been computed? (as done in [1] for instance)
- Experimental results. In my experience, these kinds of results are usually very sensitive to the number of inner steps to solve the optimization problem. In particular, if the estimation of the inner problem is crude, the value of the hyper-gradient can be poorly estimated. Moreover, to my knowledge,  quasi-second-order methods, combined with approximated gradients, can yield poor results in practice. I did not see this crucial hyper-hyper-parameter reported.

- Figure 1: could you detail how exactly this "relative eigenvalue" metric is computed? It seems crucial to make the point of the authors but it is not detailed. I guess this is the ratio of the eigenvalues over the maximal eigenvalue.

- Section 4. Could you write exactly the maths of each bilevel problem which is solved? At least in the appendix?

- I would love to see if the proposed algorithm can improve out-of-distribution generalization in meta-learning settings, like [2])

[1] Deledalle, C.A., Vaiter, S., Fadili, J. and Peyré, G., 2014. Stein Unbiased GrAdient estimator of the Risk (SUGAR) for multiple parameter selection. SIAM Journal on Imaging Sciences,

[2] Lee, K., Maji, S., Ravichandran, A. and Soatto, S., 2019. Meta-learning with differentiable convex optimization. In Proceedings of the IEEE/CVF conference on computer vision and pattern recognition (pp. 10657-10665).

Typos:
- Figure 1 eigen value >> eigenvalue

---

> ### Author Rebuttal · Authors · 2023-08-08
>
> Thank you for your comments that our idea is interesting. Here are our responses to the weakness and questions.
>
> ### Weakness
>
> > W1: A lot of statements are not properly backed.
>
> A1: The proposed method is backed by (i) theoretical analysis, which demonstrate that it can converge faster than existing gradient-based methods and converge to second-order approximate stationary points; and (ii) empirical results, where experiments on both synthetic and real data sets demonstrate its superior performance compared to existing hyper-parameter optimization methods. We are also willing to provide more support for any statements in our paper.
>
> > W2: I am not sure to properly understand the interest of the proposed method.
>
> A2: The proposed method mainly targets on hyper-parameter optimization problems where the hyper-parameters are discrete or performance metrics are non-differentiable. Please also see our response to Q1 in common responses for the major contributions of our work.
>
> ### Questions
>
> > Q3: References for statement: "These methods [Bayesian optimization, Reinforcement learning] can easily get trapped in saddle points with poor performance"
>
> A3: Thank you for pointing this out. The problem of stuck in saddle points should be only for gradient-based reinforcement learning. We will update this part in the final version to avoid such ambiguity.
>
> > Q4: What is the point of Lemma 3.1?
>
> A4: Initially, the upper-level objective of hyper-parameter optimization problems is to find the best hyper-parameter $z$ (i.e. optimize $z$). In Eq (1), we transform the upper level objective to optimize $\theta$ instead of $z$. The point of Lemma 3.1 is to justify that such transformation is reasonable, as optimizing $\theta$ is equivalent to optimizing $z$. We will make this point more clear in the final version.
>
> > Q5: Definition of what is called a saddle point could be recalled (i.e. negative eigenvalues of the hessian?)
>
> A5: Thank you for mentioning this. We will add this definition next to the definition for second-order stationary point in the final version.
>
> > Q6:  Compare to l-BFGS
>
> A6: We have added experiments using L-BFGS and include it in the revised figure. Please check the PDF file in general response. The performance of L-BFGS is similar to Newton's method in our experiments. We will also add the revised figures to the final version.
>
> > Q7: Experimental results. In my experience, these kinds of results are usually very sensitive to the number of inner steps to solve the optimization problem. In particular, if the estimation of the inner problem is crude, the value of the hyper-gradient can be poorly estimated. Moreover, to my knowledge, quasi-second-order methods, combined with approximated gradients, can yield poor results in practice. I did not see this crucial hyper-hyper-parameter reported.
>
> A7: For experiment on synthetic data, the number of inner steps is set to 40 by default, and we compare the impact of using different number of inner steps in Section 4.3. Results demonstrate that only using few steps in inner loop yields poorly estimated hyper-gradients and worse final performance. When the number of inner steps are too large, it leads to huge computation cost. We will add more descriptions about the number of inner steps in real-world experiments in final version.
>
> > Q8: Figure 1: could you detail how exactly this "relative eigenvalue" metric is computed? It seems crucial to make the point of the authors but it is not detailed. I guess this is the ratio of the eigenvalues over the maximal eigenvalue.
>
> A8: Thank you for pointing this out. Your understanding is correct. "Relative eigenvalue" refers to the ratio of eigenvalues over the max eigenvalue. We will add more descriptions about it in the final version.
>
> > Q9: Section 4. Could you write exactly the maths of each bilevel problem which is solved? At least in the appendix?
>
> A9: Here we give the mathematical formulation for each problem in our experiments:
>
> (1) For experiment on synthetic data, the mathematical formulation can be referred to Eq (1) in Section 3.1, where $z$ corresponds to feature mask,  $p_{\theta}$ uses the sigmoid function and details can be found in Section 4.1. We use AUC as the validation evaluation metric.
>
> (2) For hyper-parameter optimization for knowledge graph completion, the mathematical formulation can also be referred to Eq (1), where $z$ corresponds to a set of categorical hyper-parameters (details in Table 1), $p_\{\theta\}$ uses the softmax-like function as defined in Section 4.2.1. We use MRR as the validation evaluation metric.
>
> (3) For schedule search on learning with noisy training data, the mathematical formulation can be writen as: $\theta^*=\arg \min_{\theta} J(\theta), s.t. w^*(z)=\arg \min_{w} L_{tra}(w,R_z)$, where $J(\theta)=E_{z \sim p_{\theta}(z)} M_{val}(w^*(R_z))$. $R_z$ is the schedule function parameterized by Eq (7) in Section 4.2.2, and $z$ is the corresponding hyperparameter. For $p_\{\theta\}$, we use Beta distribution for each element of $\beta$ in Eq (7) and use Dirichlet distribution for $\alpha$ in Eq (7). The validation evaluation metric is accuracy.
>
> We will add these formulations for each problem to Appendix in final version.
>
> > Q10: I would love to see if the proposed algorithm can improve out-of-distribution generalization in meta-learning settings, like [2]
>
> A10: Thank you for raising this interesting application. As far as we know, most meta-learning problems deal with differentiable evaluation metrics and continuous hyper-parameters (e.g., model initialization). Therefore, they are not the main focus of our method. We will see if there are some problems in out-of-distribution generalization with non-differentiable metrics or discrete hyper-parameters, where we believe the proposed method should also lead to improvements.
>
> > Q11: Writing and typos.
>
> A11: Thank you for mentioning these errors. We will revise all of them in the final version of the paper.

---

> > ### Comment · Reviewer_EwuD · 2023-08-21
> > **Read the rebuttal**
> >
> > I thank the authors for the clarifications, I still think this is a good paper and should be accepted, I will keep my score unchanged

---

> > > ### Author Response · Authors · 2023-08-22
> > > **Thanks!**
> > >
> > > Thank you for your positive comments on our paper. We are also willing to engage in more discussions if you have any other questions.

---

### Official Review · Reviewer_gbiM · 2023-07-07

**Soundness:** 2 fair
**Presentation:** 3 good
**Contribution:** 2 fair
**Rating:** 7
**Confidence:** 2

**Summary:**

This paper studies the problem of hyper-parameter optimization in the context of machine learning. Through stochastic relaxation, the problem can be formulated as bilevel optimization, and the authors propose to use cubic regularization to solve the optimization problem. It is shown that under some regularization conditions on the loss function and hyper-parameter distribution, the algorithm efficiently converges to an approximate second-order stationary point. Furthermore, experiments are conducted to verify the effectiveness of the proposed method.

**Strengths:**

1. The paper is well-written and quite easy to read. The authors explain the motivations, method and results in a very clear way.
2. Extensive experiments are conducted to demonstrate the wide applicability of the proposed method. All experiment settings are explained in details.

**Weaknesses:**

While the authors provide theoretical guarantees for the proposed method, it seems that these guarantees can be directly derived from known results on the convergence of cubic regularization methods. At the same time, because the problem considered is bilevel, some assumption seems less natural. For example, the Lipschitz-Hessian assumption on $J(\theta)$ is quite difficult to verify in practice, since the expression of $J$ itself contains a minimization problem. I wonder if it is possible to obtain convergence results under more 'fundamental' assumptions.

**Questions:**

1. In Assumption 3.5, the estimation of gradient and Hessian may be biased. How does this affect the final convergence rate?
2. How does the proposed method compared with other methods that are not based on stochastic relaxation?

**Limitations:**

The authors have adequately addressed the limitations and potential societal impact of their work.

---

> ### Author Rebuttal · Authors · 2023-08-08
>
> Thank you for your comments for both the strength and weakness on our paper. We would like to first answer the questions raised in the weakness part:
>
> > W1: While the authors provide theoretical guarantees for the proposed method, it seems that these guarantees can be directly derived from known results on the convergence of cubic regularization methods.
>
> A1: It is not trivial to obtain our theoretical analysis from known results on the convergence of cubic regularization methods.
> Since we are solving a bi-level optimization problem, the gradient and Hessian used in the cubic regularization methods may be inexact. As such, we need to bound the error of estimated gradient and Hessian from the solution to lower-level problem, which is different from any known results to the best of our knowledge. We will make the technical novelty more clear in our next version.
>
> > W2: some assumption seems less natural. For example, the Lipschitz-Hessian assumption on $J(\theta)$ is quite difficult to verify in practice, since the expression of $J$ itself contains a minimization problem. I wonder if it is possible to obtain convergence results under more 'fundamental' assumptions.
>
> A2: Thank you for your suggestion. Recall that the definition of Lipshitz Hessian is $|| \nabla^2 J(\theta) - \nabla^2 J(\phi) || \le \rho || \theta -\phi ||$ for any $\theta$ and $\phi$, and from Proposition 3.3, we have
> $\nabla^2 J(\theta)  =  \int H^*(\theta, z) p_{\theta}(z) dz$,
> where $H^*(\theta, z)$ depends on the objective value, $\nabla \log p_{\theta}(z)$ and $\nabla^2 \log p_{\theta}(z)$.
> Then the assumption of Lipschitz Hessian can be deduced by assuming 1) bounded objective value; and 2) bounded differences of $\nabla \log p_{\theta}(z)$ and $\nabla^2 \log p_{\theta}(z)$ with different $\theta$, where similar assumptions are made in Assumption 3.5(ii). We will consider changing to more "fundamental" assumptions in the next version.
>
> ### Answers to remaining questions
>
> > Q3: In Assumption 3.5, the estimation of gradient and Hessian may be biased. How does this affect the final convergence rate?
>
> A3: In our proof, the estimation errors of gradient and Hessian do affect the convergence rate, and inaccurate gradient or Hessian slow down convergence. Nevertheless, the dependency is very complex, and we consider the worst case performance in theorem 3.6 to make it simple.
>
> > Q4: How does the proposed method compared with other methods that are not based on stochastic relaxation?
>
> A4: In Section 4.2 (Experiments on Real-world Data), we compare the proposed method with other methods that are not based on stochastic relaxation, including random search, Bayesian optimization, Hyperband and reinforcement learning. In the two applications, the proposed method achieves the best performance.

---

> > ### Comment · Reviewer_gbiM · 2023-08-18
> > **Response**
> >
> > Thank you for the clear response to my questions which have adequately addressed my concerns. I will keep my rating and still recommend accept.

---

> > > ### Author Response · Authors · 2023-08-18
> > > **Thanks!**
> > >
> > > Thank you for your questions and your affirmation of our paper. We are also willing to engage in more discussions if you have any other questions.

---

### Official Review · Reviewer_S7HE · 2023-07-19

**Soundness:** 3 good
**Presentation:** 3 good
**Contribution:** 3 good
**Rating:** 6
**Confidence:** 4

**Summary:**

This paper proposes a new optimization based technique for hyper-parameter tunings using Adaptive Cubic Regularized Newton method (ARC) based on stochastic relaxation. The author highlights the limitations of the existing hyper-parameter optimization algorithms.They show that their suggested method achieves better convergence guarantees as existing methods only prove convergence to first order methods where their work also achieves second order optimality conditions. The authors also run some numerical experiments to show that the algorithm outperforms existing methods in terms of faster convergence.

**Strengths:**

* A weakness of existing methods is clearly identified and solved.
* The originality lies in being the first to use the second order method (ARC) in a bi-level optimization problem for hyper-parameter tuning.
* The paper is well written and the result can serve as the base of future work in  using second order methods for hyper-parameter tunings.
* Some minimal numerical results are shown


**Weaknesses:**

* The major contribution of this work should be explicitly stated in a section called “Our contributions”.
* The authors use ' the curse of dimensionality' as an example of a disadvantage for existing methods; however, when explaining their proposed approach and arguing that even they have to compute the full Hessian, this will not be an expensive operation since hyper-parameters are often low dimensional. This contradicts with the disadvantage mentioned earlier.
* The authors didn’t compare their results against the global optimal (if exists) set of hyper-parameters in their numerical experiments.
* In Algorithm.1, the authors didn’t explain how to optimize the lower-level objective in (1)
* In Algorithm.1, the authors didn’t explain if ARC is being solved using an iterative approach or a factorization based approach when solving the subproblems.
* In their analysis, the authors are not taking into account the cost of solving the ARC subproblems.
* For the synthetic data experiments, the authors didn’t use any reference and they didn’t explain why they didn't use real data for the problem of filtering useful features.
* The details of the experiment parts are not enough so that people can reproduce the results. For example, the authors don't state what machine learning models they used.
* The proof is a bit tricky and it has some typos and errors.

**Questions:**

* A typo under Table 1 should be fixed.
* In all figures, the x-axis is confusing. What do you mean by number of trained models? Also does this mean that the AUC on the y-axis is computed by averaging across all the trained models?
* What machine learning model you used for all the experiments. More details are needed for example for the classification did you use Decision Tree, SVM, Logistic regression …. Or in the image setup, what neural network models you used?
* In figure 4, you can add (M) next to the upper loop iterations so that it is easier to understand the plot and what parameters are you considering in your ablation study. Also add the unit for the relative training time.
* The details about the experiment are important as reproducibility of the work is a major concern. Even if you don’t add a new dataset or create a new machine learning model, there is still a need to reproduce your results and validate that.
* The details of the experiments should include: what ML models are being trained, the environment, how to set up the datasets and access them, which packages are being used, ARC solver package ….
* The code of your work also should be attached to the submission and later in your final paper, it is recommended to add the github repository link.
* Also in your code, did you fix the seed so that people can get similar results as what you got.

Comments about the proof:
* I checked the proof carefully and most of the math seems correct, however following the proof is a bit tricky and I would suggest it if you can make it more clear. Also, if you can give more explanation about the intuition behind those lemmas.
* Some of the lemmas better to be referenced as facts since you are using the results of previous works: Lemma B.1, Lemma B.2, Lemma B.4.
* In Lemma B.1 if you can refer to the Hessian as $\rho$-Lipschitz.
* Prerequisite of Theorem 3.6 should appear on its own as an equation because you are referring to it in multiple places.
* In the proof of Preposition 3.2, there is a typo. There should be $\grad \log$ in the last two lines of the proof.
* In Lemma B.3, if you can elaborate more the steps as it is not clear how you get rid of the summation term that appears in the definition of the approximate gradient $g^m$. Since the expectation is with respect to the probability $p_\theta$ and your summation is in term of $1/k$, but the $p_\theta$ isn’t uniform so to interchange those terms.
* In Lemma B.3, how did you get rid of the $\min$ term that appears in the prerequisite of Theorem 3.6 since you only used the first term in from $\min (term_1, term_2)$.
* In Lemma B.5, similar comment to Lemma B.3. How you get rid of the summation term that appears in the definition of the approximate hessian. Also, you only used the second term that appears in the $\min(term_1, term_2)$ from the prerequisite of Theorem 3.6.
* In Lemma B.5, in your first line of proof you say “By definition, we have“, but you didn’t refer to any definition.
* In Lemma B.5, I guess you have a typo in the equality that involves $K$. I guess, this should be $b_2 ^ 2$ instead of $c_2 ^ 2$.
* In Lemma B.5, I guess you have a typo in the inequality bounding the norm of the difference between the approximate hessian and the actual hessian. $\|| B ^ m - \grad J(\theta ^ m)\||$ should be  $\|| B ^ m - \grad ^ 2 J(\theta ^ m)\||$. Also in this lemma, you use the l-2 norm and the spectral norm interchangeably but it is not clear if the math is adjusted at all steps. Or you have some typos.
* Lemma B.6 is well known facts about ARC, so you can refer to some citations and in that case there is no need to do the proof.
* In Lemma B.7, when you refer to both Lemmas B.3 and B.5 you mention the inequalities from those lemmas, but without taking into consideration that each one satisfies the inequality with a different probability.
* In lemma B.9, similar comment to the above when referring to lemmas B.3 and B.5.
* In lemma B.9, the inequality that appears after the line “And from Lemma B.5, we will have” seems wrong since as I understand you are using Cauchy Schwarz inequality here and it is not clear why you have $\||\delta ^ m\|| ^ 2$ and from where the half term appears. I guess the correct term should be $c_2 \sqrt{\rho \epsilon} \|| \delta ^ m \||$. Then the proof needs to be updated.
* In Lemma B.9, also it is not clear from where you got the result in case 2 that $\tilde c(\delta ^ m)$  <= -\frac{1}{96} \sqrt {\epsilon ^ 3 / \rho}$.


**Limitations:**

Exposed by the authors.

---

> ### Author Rebuttal · Authors · 2023-08-08
>
> Thank you for your comments on both strength and weakness of our paper. Here are our responses to your comments on weakness and questions:
>
> ## Weakness
>
> > W1: The major contribution of this work should be explicitly stated in a section called "Our contributions".
>
> A1: Please see our response to Q1 in the general response part.
>
> > W2: The authors use 'the curse of dimensionality' as an example of a disadvantage for existing methods; however, when explaining their proposed approach and arguing that even they have to compute the full Hessian, this will not be an expensive operation since hyper-parameters are often low dimensional. This contradicts with the disadvantage mentioned earlier.
>
> A2: Thank you for pointing this out. As the hyper-parameters are often low-dimensional, it may not be appropriate to refer to "curse of dimension" for a disadvantage of existing methods. Nevertheless, even with low-dimensional hyper-parameters, existing methods still suffer from slow convergence, as is also demonstrated in experiment part. We will revise this part to avoid such contradictions in final version.
>
> > W3: Compare against global optimal set of hyper-parameters in experiments.
>
> A3: For experiments on real-world data, it is hard to obtain the global optimal set of hyper-parameters in advance, so here we only consider the experiment on synthetic data, where the global optimal set of hyper-parameters will be masking all the noise and retaining all the useful features. We have added its performance to the revised Figure 1(a) and you could check it in the PDF file in general response part. From the revised figure, we can see that the best validation AUC obtained by the proposed method is close to global optimum, while all the other baseline methods have much larger gaps.
>
> > W4: In Algorithm.1, the authors didn’t explain how to optimize the lower-level objective in (1).
>
> A4: Our proposed method should be compatible with any optimizer for lower-level objective, e.g. SGD, Adam or AdamW. In our experiments, we use AdamW in the experiment on synthetic data and use Adam in experiments on real-world data. We will make this more clear in the final version.
>
> > W5: In Algorithm.1, the authors didn’t explain if ARC is being solved using an iterative approach or a factorization based approach when solving the subproblems.
>
> A5: In Algorithm 1, ARC is solved using an iterative approach, similar to the method described in section 6 in [1]. We will make this more clear in the final version.
>
> > W6: In their analysis, the authors are not taking into account the cost of solving the ARC subproblems.
>
> A6: Thank you for raising this question. In our experiments, the number of hyper-parameters is often much less than 100, while the models in lower-level problems have millions of parameters. As such, it should take much longer time to solve lower-level problems than solve the ARC subproblems. It is also empiricaly verified in Section 4.4 that the time cost of updating $\theta$ (which includes the time cost of solving the ARC subproblems) is negligible in most cases. Therefore, we focus more on the time cost on solving lower-level problems, rather than solving the ARC subproblems.
>
> > W7: For the synthetic data experiments, the authors didn’t use any reference and they didn’t explain why they didn't use real data for the problem of filtering useful features.
>
> A7: We will add citations on feature selection problem in final version, e.g. [2], [3]. As we already have experiment for real data in Section 4.2, the use of synthetic data also enables us to compare against global optimal, which may be hard to obtain if we use real-world data.
>
> > W8: Experiment details
>
> A8: Please see our response to Q3 in common responses.
>
> > W9: Typos and errors in proof
>
> A9: We will thoroughly revise our paper and appendix in our final version.
>
> [1] Adaptive cubic regularisation methods for unconstrained optimization. Part I: motivation, convergence and numerical results. Mathematical Programming, 2011.
>
> [2] An introduction to variable and feature selection. JMLR, 2003.
>
> [3] Feature selection using stochastic gates. ICML 2020
>
> ## Questions
>
> > Q10: Typo under Table 1
>
> A10: Thank you for pointing this out. We will revise it in the final version.
>
> > Q11: In all figures, the x-axis is confusing. What do you mean by number of trained models? Also does this mean that the AUC on the y-axis is computed by averaging across all the trained models?
>
> A11: Thank you for pointing this out. We have changed the x-axis to running time and you could check the revised figures in the PDF file in general responses. The y-axis plot the best validation performances obtained at that time. We will make this more clear in next version.
>
> > Q12: Experiment details
>
> A12: Please see our response to Q3 in common responses.
>
> > Q13: In figure 4, you can add (M) next to the upper loop iterations so that it is easier to understand the plot and what parameters are you considering in your ablation study. Also add the unit for the relative training time.
>
> A13: Thank you for your suggestions. We have uploaded revised figures on running time in the PDF file in general response. We will also add notations to figures on upper loop iterations.
>
> > Q14: The code of your work also should be attached to the submission and later in your final paper, it is recommended to add the github repository link.
>
> A14: Thank you for your suggestion. The code still needs some clean up to make it easier to use. We will make our code public on GitHub along with our final version.
>
> > Q15: Random seed in our code
>
> A15: We have fixed the seeds in all our experiments. We will explicitly mention this in the next version of our paper.
>
> > Q16: Comments about the proof.
>
> A16: Thank you for your thorough comments. We have checked our proof and found that these typos/errors did not affect the correctness of key theoretical results. We will make all these revisions in final version.

---

> > ### Author Response · Authors · 2023-08-10
> > **Responses to comments on our proof**
> >
> > We would like to thank reviewer S7HE again for detailed comments to our proof. We have thoroughly checked our proof and did not find critical issues that will affect its correctness. Here are our responses to these comments:
> >
> > > Q17: I checked the proof carefully and most of the math seems correct, however following the proof is a bit tricky and I would suggest it if you can make it more clear. Also, if you can give more explanation about the intuition behind those lemmas.
> >
> > A17: Thank you for your suggestions. We will try to revise the proofs to make them more clear. We will also add more explanations to each lemma used in the proof in final version.
> >
> > > Q18: Some of the lemmas better to be referenced as facts since you are using the results of previous works: Lemma B.1, Lemma B.2, Lemma B.4. \
> > Lemma B.6 is well known facts about ARC, so you can refer to some citations and in that case there is no need to do the proof.
> >
> > A18: While these lemmas are really known facts in previous work, some people may not be familiar with them. As such, we suppose it may be better to also include them in appendix. We will also add references to them in final version.
> >
> > > Q19: In Lemma B.1 if you can refer to the Hessian as $\rho$-Lipschitz.
> >
> > A19: Thank you for your suggestion. We will add it in final version.
> >
> > > Q20: Prerequisite of Theorem 3.6 should appear on its own as an equation because you are referring to it in multiple places.
> >
> > A20: The conditions of Theorem 3.6 directly impact the performance of final solution ($\epsilon$). As such, we suppose it would be better to put it as a prerequisite of Theorem 3.6 rather than a separate assumption.
> >
> > > Q21: In the proof of Preposition 3.2, there is a typo. There should be \grad log in the last two lines of the proof.
> >
> > A21: Thank you for pointing this out. We will revise it in final version.
> >
> > > Q22: In Lemma B.3, if you can elaborate more the steps as it is not clear how you get rid of the summation term that appears in the definition of the approximate gradient $g^m$. Since the expectation is with respect to the probability $p_\theta$ and your summation is in term of 1/k, but the $p_\theta$ isn’t uniform so to interchange those terms... In Lemma B.5, similar comment to Lemma B.3. How you get rid of the summation term that appears in the definition of the approximate hessian.
> >
> > A22: Thank you for pointing this out. Since each $z^k$ is an i.i.d. sample from probability distribution $p_\theta$, the expectation will not change, and we can get rid of the summation (actually average) term. We will revise this part to make it more clear in final version.
> >
> > > Q23: In Lemma B.3, how did you get rid of the min term that appears in the prerequisite of Theorem 3.6 since you only used the first term in from min(term1, term2) ... Also, you only used the second term that appears in the min(term1, term2) from the prerequisite of Theorem 3.6.
> >
> > A23: Thank you for pointing this out. Obviously, we have min(term1, term2) smaller than both term1 and term2, so we can simply replace min(term1, term2) by either term1 or term2 for an upper bound. We will revise this part to avoid such ambiguity.
> >
> > > Q24: In Lemma B.5, in your first line of proof you say “By definition, we have“, but you didn’t refer to any definition.
> >
> > A24: Here we are referring to the definition for spectral norm. We will add this to final version.
> >
> > > Q25: In Lemma B.5, I guess you have a typo in the equality that involves K. I guess, this should be $b_2^2$ instead of $c_2^2$.
> >
> > A25: Yes, it should be $b_2^2$. We will update it in final version.
> >
> > > Q26: In Lemma B.5, I guess you have a typo in the inequality bounding the norm of the difference between the approximate hessian and the actual hessian. $|| B^m-\nabla J(\theta) ||$ should be $|| B^m-\nabla^2 J(\theta) ||$. Also in this lemma, you use the l-2 norm and the spectral norm interchangeably but it is not clear if the math is adjusted at all steps. Or you have some typos.
> >
> > A26: Thank you for pointing this out. In Lemma B.5, we always use spectral norm as distance metric for matrices. We have checked the proof to avoid ambiguity of using two different norms for different objects (l2-norm for vectors and spectral norm for matrices). We will also revise the error in $|| B^m-\nabla^2 J(\theta) ||$

---

> > > ### Author Response · Authors · 2023-08-10
> > > **Responses to comments on our proof (cont.)**
> > >
> > > > Q27: In Lemma B.7, when you refer to both Lemmas B.3 and B.5 you mention the inequalities from those lemmas, but without taking into consideration that each one satisfies the inequality with a different probability; In lemma B.9, similar comment to the above when referring to lemmas B.3 and B.5.
> > >
> > > A27: Thank you for pointing this out. Using the same notations, consider two random events A and B that happens with probability at least $1-\delta_1$ and $1-\delta_2$, respectively. Then the probability that either of them does not happen is at most $\delta_1+\delta_2$, hence the probability that both of them happen is at least $1-(\delta_1+\delta_2)$, and can be made sufficiently large with sufficiently small $\delta_1$ and $\delta_2$. We will revise this part in final version to make it easier to understand.
> > >
> > > > Q28: In lemma B.9, the inequality that appears after the line “And from Lemma B.5, we will have” seems wrong since as I understand you are using Cauchy Schwarz inequality here and it is not clear why you have $|| \delta^m ||^2$ and from where the half term appears. I guess the correct term should be $c_2 \rho \epsilon || \delta^m ||$. Then the proof needs to be updated.
> > >
> > > A28: Thank you for pointing this out. Here we are actually bounding $\frac{1}{2} (\Delta^m)^{\top} (\nabla^2 J(\theta)-B^m) \Delta^m$, and you are correct if we would like to bound $|| (\nabla^2 J(\theta)-B^m) \Delta^m ||$. We will revise this error and the rest of proof does not need any change.
> > >
> > > > Q29: In Lemma B.9, also it is not clear from where you got the result in case 2 that $\tilde{c} (\delta^m) \le -\frac{1}{96} \sqrt {\epsilon ^ 3 / \rho}$.
> > >
> > > A29: The proof for case 2 directly uses the condition in Lemma B.9 that we assume. We will revise this part to directly refer to the condition.

---

> > > > ### Comment · Reviewer_S7HE · 2023-08-16
> > > > **Satisfactory answers**
> > > >
> > > > I appreciate the authors for their responses and the updates they have made. I am raising my score as the authors have addressed my comments.

---

> > > > > ### Author Response · Authors · 2023-08-16
> > > > > **Thanks!**
> > > > >
> > > > > We are pleased to know that our responses have addressed your previous comments. We are also willing to engage in more discussions if you have any other questions.

---

### Author Rebuttal · Authors · 2023-08-08

We thank all reviewers for their constructive feedback. Here we colloct some common questions and reply them in this general response. A PDF file containing all revised figures is also uploaded.

> Q1: major contributions

A1: The major contributions of this work include:

(1) We propose to utilize cubic regularization to accelerate convergence and avoid saddle points in hyper-parameter optimization problems.

(2) We provide theoretical analysis of the proposed method, showing that the proposed method converge to approximate second-order stationary points with inexactly solved lower level objective considered.

(3) We verfied the effectiveness of the proposed method in experiment on synthetic and real-world data.

We will also update these contributions to a separate "Our contributions" section in the final version of the paper.

> Q2: typos and errors

A2: Thank you for pointing these out. We will make thorough revisions in final version.

> Q3: experiment details

A3: Thank you for your suggestions. Most details on our experiments can be found in appendix. Here we present some details for easy reference:

(1) ML model. For experiment on synthetic data, we use a linear model with single layer as feature classifier. For hyper-parameter optimization for knowledge graph completion, we use traditional embedding based model, e.g., TransE, RotatE, ComplEx (see Table 1). For schedule search on learning with noisy training data, we use a 5-layer CNN similar to LeNet as the model for image classification. We will update more details to revised appendix in final version.

(2) Environment. For all experiments, we use Python 3.7, PyTorch 1.13 and torchvision 0.14. Installing these packages should be enough to replicate all our experiments.

(3) Datasets. For synthetic data, the generation process is described in Line 232-236 in Section 4.1: "We construct a synthetic dataset on 5-way classification, where the inputs are 50-dimensional vectors. Of the 50 features, 25 have different distributions based on their ground-truth labels, generated from Gaussian distributions with different means for each class and the same variance. The remaining 25 features are filled with i.i.d. Gaussian white noise."
For knowledge graph completion, the two data sets, FB15k-237 and WN18RR, are well-known in the field of KG.
For learning with noisy labels, the CIFAR-10 data set is also well-known and can be easily accessed from torchvision package.

(4) Packages. Despite standard packages for deep learning (e.g. PyTorch), we do not use extra packages (e.g. ARC solver packages) in our experiment.

We will also update more details to revised appendix in final version.

> Q4: revise figures

A4: We have uploaded the revised figures in the attached PDF file. Our modifications include:

(1) Add the performance global optimal in Figure 1(a) for easy reference

(2) Add the result of BFGS in Figure 1

(3) Adjust descriptions of axes in all figures

---

> ### Comment · Area_Chair_67R8 · 2023-08-19
> **Clarification request**
>
> Dear authors,
>
> I was reading the reviews and the paper. One thing that was not clear to me is that in the experiments (Figure 2 & 4 in the submitted paper) whether the baselines BO, random, RL and hyperband are applied to stochastic relaxation or the original hyperparameter search problem (e.g., Table 1). I ask this because many of these methods can be directly applied to discrete hyperparameter search problems but gradient based methods cannot. Could you clarify this point? This is not clear to me based on the text.
>
> Sorry about the last minute request.
>
> Thanks,
> Your area chair

---

> > ### Author Response · Authors · 2023-08-20
> > **Clarification on our experiments**
> >
> > Dear AC
> >
> > Thank you for your question. These baseline methods are applied to the original hyper-parameter search problem. We will explicitly mention this point in our camera ready version.
> >
> > Feel free to ask any further questions if you find any other unclear points.
> >
> > Best,
> >
> > Authors

---

### Decision · Program_Chairs · 2023-09-21

**Decision:**

Accept (poster)

**Comment:**

The authors propose using cubic regularization to solve stochastic relaxations of hyperparameter optimization problems. This is not a standard application of cubic regularization because they apply it to a bilevel optimization problem where the lower-level is inexactly solved. Experimental results indicate that the approach is promising.